# Conserved domains and structural motifs that differentiate closely related Rex1 and Rex3 DEDDh exoribonucleases are required for their function in yeast

Peter W. Daniels[iD][�ław], Sophie Kelly[�], Iwan J. Tebbs[¤a☏ław], Phil Mitchell[iD]*

School of Biosciences, The University of Sheffield, Sheffield, United Kingdom

☏ These authors contributed equally to this work
¤a Current address: Stem Cell Donor Registry, NHS Blood and Transplant, Bristol, UK
* p.j.mitchell@sheffield.ac.uk

## Abstract

The DEDD family of exonucleases has expanded through evolution whilst retaining a conserved catalytic domain. One subgroup with closely related catalytic DEDD domain sequences includes the yeast enzymes Rex1 (RNA exonuclease 1) and Rex3, the metazoan REXO1 (RNA exonuclease 1 homologue) and Rexo5 proteins, and the plant protein Sdn5 (small RNA degrading nuclease). Comparison of protein structure models and sequence analyses revealed that this group can be differentiated into two distinct clades consisting of Rex1, Rexo5 and Sdn5 on the one hand, and Rex3 and REXO1 on the other. The catalytic domain of Rex1-related proteins is inserted within a conserved, discontinuous alkaline phosphatase (AlkP) domain. The AlkP domain of yeast Rex1 contains three surface loops that are modelled to be directed towards the DEDD domain, one of which forms an extended helical arch that is found in homologues across fungi and plants. We show that this arch and an adjacent loop are required for Rex1-mediated processing of 5S rRNA and tRNA in *Saccharomyces cerevisiae*. Rex3-related proteins, including REXO1, lack the AlkP domain but contain a KIX domain (CREB kinase-inducible domain (KID) interacting domain) and a cysteine- and histidine-rich domain (CHORD) adjacent to a C-terminal DEDD domain. Deletion of the N-terminal region within yeast Rex3 spanning the KIX domain blocked its function in RNase MRP processing. In contrast to Rex1, Rex3 proteins are found in metazoans and fungi but not in plants or algae. This work identifies evolutionarily conserved structural hallmarks within Rex1 and Rex3 proteins and demonstrates that specific features are required for Rex1- and Rex3-mediated RNA processing pathways *in vivo*.

**Data availability statement:** All relevant data are within the manuscript and its supporting information files.

**Funding:** P.D. was supported by the Biotechnology and Biological Sciences Research Council (BBSRC)(https://www.ukri.org/councils/bbsrc) through the White Rose doctoral training programme and the BBSRC grant BB/V00722X/1. The sponsor played no role in the study design, data collection/analysis, decision to publish or manuscript preparation. S.K. and I.T. were undergraduate students on the MBiolSci programmes within the University of Sheffield.

**Competing interests:** The authors have declared that no competing interests exist.

## Introduction

The DEDD family of 3' exonucleases is a group of diverse enzymes that hydrolytically remove (deoxy)ribonucleotide 5'-monophosphates from the 3' end of DNA or RNA. They are found throughout biology and have a broad range of actions in the key biological processes of DNA replication and repair, gene expression and RNA turnover. The catalytic centres of these enzymes contain four conserved acidic residues (hence the term DEDD) that are located within three conserved sequence motifs known as the Exo I, Exo II and Exo III motifs [1]. The DEDD family is split into DEDDh and DEDDy subfamilies that contain either a conserved histidine or tyrosine residue within the Exo III motif. The DEDD catalytic domain is found in both exonucleases and in polymerases that have associated proof-reading activities [1,2].

The *Escherichia coli* genome encodes two DEDDh family exoribonucleases, there are eight encoded within the genome of the budding yeast *Saccharomyces cerevisiae* and fourteen expressed in human cells. These enzymes show specificity for some substrates while functioning redundantly in the processing or degradation of others. For example, the yeast enzymes Rex1–3 have overlapping yet distinct roles in the processing of noncoding RNA transcripts to generate functional, stably expressed RNAs [3]. Rex1 (RNA exonuclease 1) has specific roles in the 3' end maturation of 5S rRNA, 25S rRNA, signal recognition particle (SRP) RNA and some tRNAs [3–9]. Rex3 is specifically required for correct 3' end processing of the RNA subunit of RNase MRP [3] and functions together with Rex2 in the autoregulated degradation of *RTR1* mRNA transcripts [10]. None of the yeast DEDDh exonucleases are essential for mitotic growth but *rex1* null alleles are synthetic lethal with null alleles of the *RRP6* gene [3,11] that encodes a DEDDy family exoribonuclease or the *RRP47* gene [12] that encodes an Rrp6-associated protein [13].

One subgroup of DEDDh ribonucleases with closely related catalytic domain sequences includes yeast Rex1 and Rex3 proteins and the metazoan REXO1 and Rexo5 proteins (InterPro family IPR034922). These sequence-related proteins have some functional overlap. For example, rRNA processing events that are mediated by Rex1 in *S. cerevisiae* are carried out by Rexo5 in *Drosophila melanogaster* [14]. Humans and mice also express a sequence homologue of the Rexo5 fly protein (known as NEF-sp or REXO5) [15] but its role in the processing of noncoding transcripts to stable RNAs has not been investigated. However, yeast Rex1 and the fly Rexo5 protein lack the RNA recognition motif (RRM) domains observed in vertebrate REXO5 proteins [14] that are required for its recently reported role in DNA repair [16]. Moreover, another human protein called REXO1 (RNA exonuclease 1 homologue) [17] has not been functionally characterised. The functional and structural relationships between members of the Rex1/Rex3 family of proteins are therefore currently unclear.

In an earlier study we showed that the predicted structural architecture of yeast Rex1 is very closely related to that of the uncharacterised proteins Yfe9 from *Schizosaccharomyces pombe* and Sdn5 (Small RNA degrading nuclease 5) from *Arabidopsis thaliana* [18]. The N- and C-terminal regions of these proteins are predicted to

constitute a discrete discontinuous domain that is related to the alkaline phosphatase (AlkP) superfamily and that we have shown to be required for the stable expression and catalytic activity of Rex1 [18]. Whether this dual domain structural organisation is more broadly conserved in Rex1-related proteins across eukaryotes, and whether or not structural features within this domain are required for Rex1 function, have not yet been reported.

We have carried out phylogenetic sequence analyses of Rex1 and Rex3 exonucleases and compared high confidence AlphaFold (AF) models retrieved from the AF protein structure database, with the aim of identifying structural characteristics of each enzyme. We report that the AlkP domain is a common feature of Rex1, Rexo5 and Sdn5 proteins that are found in all major eukaryotic lineages. Mutagenesis studies reveal that insertions on the surface of the AlkP domain of Rex1, including the highly conserved helical "arch" that is modelled to be directed towards the DEDDh domain, are required for 5S rRNA and tRNA processing in *S. cerevisiae*. Furthermore, we show that REXO1 and Rex3 proteins have a common domain architecture distinct from that of Rex1, consisting of a conserved KIX (CREB kinase-inducible domain (KID) interacting domain) domain and a single CHORD (cysteine- and histidine-rich domain) domain [19–21], in addition to the C-terminal DEDDh domain. Deletion analyses show that the N-terminal region of the yeast Rex3 protein, which spans the KIX domain, is required for its function in the 3' end maturation of RNase MRP RNA *in vivo*. This study identifies structural features of Rex1 and Rex3 exoribonucleases, in addition to the DEDDh catalytic domain, that are required for their function in specific RNA processing pathways.

## Results and discussion

### The AlkP domain is conserved in Rex1-related proteins

AF [22,23] predicts a high confidence dual domain structure for yeast Rex1 (a mean per reside distance difference test, abbreviated pLDDT, score of 92 for residues 53–553) consisting of a central DEDDh catalytic domain that is inserted within a domain comprising sequences from both the N- and C-terminal regions of the protein [18] (Fig 1A). This discontinuous domain has the same fold as experimentally determined structures for cofactor-independent phosphoglycerate mutases (iPGMs) (Fig 1C) and phosphopentomutases [24,25], two members of the alkaline phosphatase (AlkP) superfamily [26]. We had previously referred to this domain as the RYS domain since it was observed in Rex1, Yfe9 and Sdn5 proteins [18]. For simplicity, and given the broader evolutionary conservation of this domain (see below), we refer to this domain here as the AlkP domain of Rex1.

The core AlkP domain fold consists of a predominantly parallel, seven stranded β-sheet that is supported by α-helices on both sides (Fig 1B and C, helices and loops on the internal faces are not shown for clarity). The β-strands are numbered 1–7 in Fig 1, from the N- to C-terminus. Polypeptide sequences spanning strands 1 and 2 (coloured blue) consist of sequences from the N-terminal region of Rex1, while strands 3–7 (coloured red) are from the C-terminal region (Fig 1B). The central DEDDh domain of Rex1 is connected to the AlkP domain by flexible polypeptide linkers that lead from β2 into the DEDDh domain, and from the DEDDh domain back into β3. Prominent surface loops of the yeast Rex1 structure are referred to here as loop 1, that leads from β2 into the linker connecting the AlkP and DEDDh domains, loop 2 that leads from β5 towards β6, and loop 3 that connects β6 and β7 (Fig 1B).

Analysis of structure similarity clusters within AF [28,29] (version 4 of the AlphaFold database) revealed that the dual domain architecture of yeast Rex1 is found in proteins from fungi, flowering plants (including several Sdn5 homologues), green algae, amoebozoa, diatoms and flagellates, as well as in the metazoan organisms *Clytia hemisphaerica* (a cnidarian) and *Hirondella gigas* (an amphipod) (see S1 Table). The structure similarity clusters do not include the functionally homologous Rexo5 protein from *D. melanogaster* or the vertebrate homologues NEF-sp/REXO5 or REXO1, although reverse analysis of structure similarity clusters of the *D. melanogaster* Rexo5 protein identified Rex1 homologues in other yeasts (S1 Table). The DEDDh domain of Rexo5, like yeast Rex1, is also positioned between extensive N- and C-terminal flanking regions. Vertebrate REXO5 proteins also have a central DEDDh domain and contain two RNA recognition motif

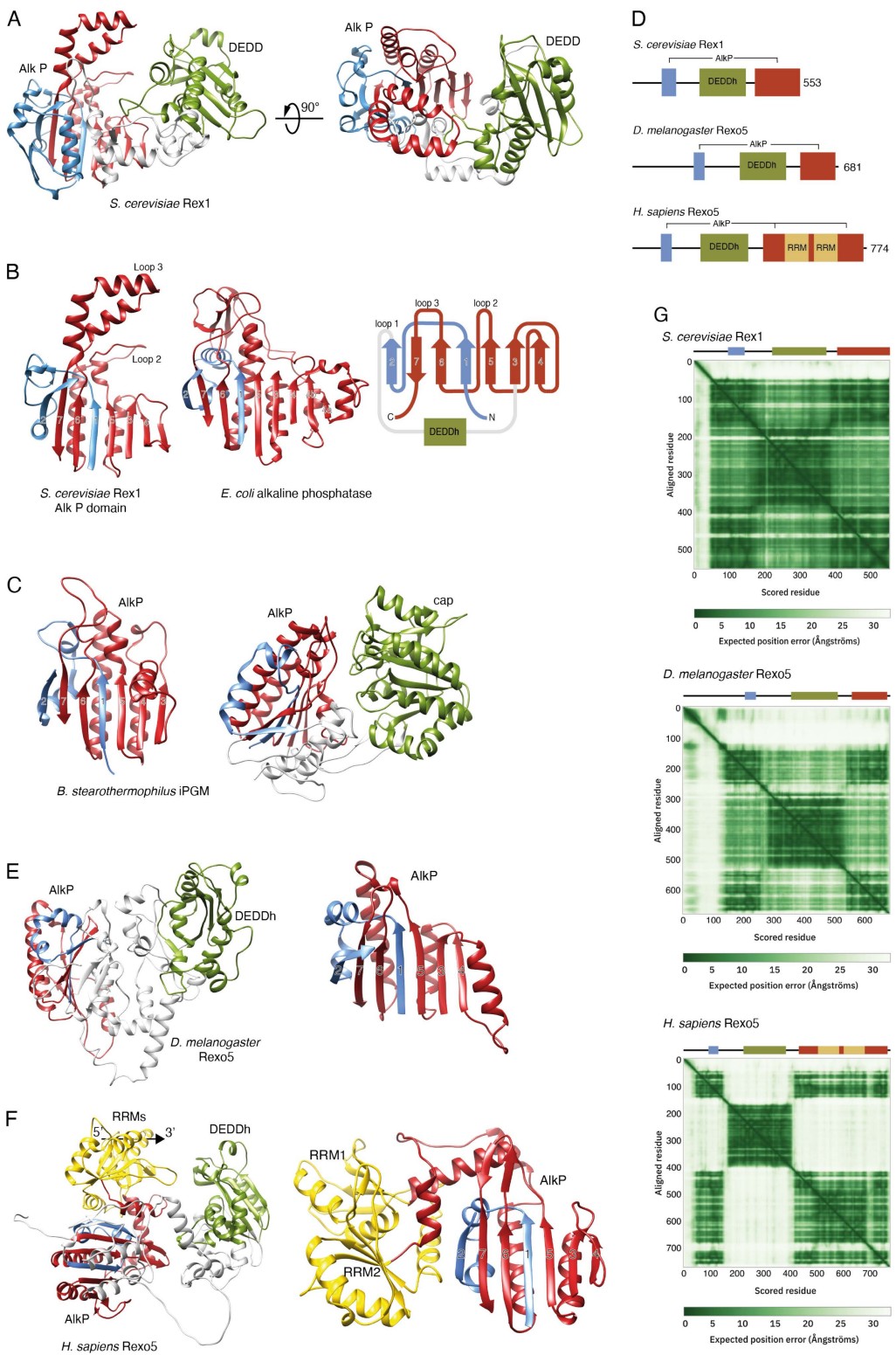

**Fig 1. Rex1 and Rexo5 orthologues have a discontinuous alkaline phosphatase fold.** Ribbon structure diagrams of Rex1- and AlkP-related proteins. The central DEDDh and cap domains of Rex1-related proteins and iPGM, respectively, are coloured green; N- and C-terminal portions of AlkP domains are coloured blue and red, respectively. The RRM domains inserted within human Rexo5 are coloured gold. (A) Ribbon structure (residues

53-553) of the yeast Rex1 AF model (AF-P53331). (B) Comparison of the yeast Rex1 AlkP domain with AlkP from *E. coli* (PDB ID 1ALK) [27]. The perspective shown is towards the surface of the AlkP domain ß-sheet. The Rex1 topology diagram shows the ß-sheet numbering, the insertion points of the DEDDh domain within the AlkP domain and the location of prominent loops. Regions before strand 1, between strands 2 and 3, and connecting strands 3 and 4 within the structure of AlkP, are omitted for clarity. (C) AlkP domain and dual domain structure of iPGM from *B. stearothermophilus* (PBD ID 1EJJ) [24]. The left-hand panel shows the surface of the ß-sheet, as in panel B. The right-hand panel shows the AlkP and cap domains. (D) Schematic, drawn to scale, of the domain structure of Rex1 and Rexo5 proteins. (E) Ribbon diagram (residues 145-681) of the AF model of fly Rexo5 (AF-Q9VRX7) and the AlkP domain alone, viewed with the perspective shown in panels B and C. (F) AF model of the human Rexo5 protein (AF-Q96IC2, residues 49-774) and the AlkP/RRM module, viewed from the same perspective as shown in panels B, C and **E.** The inferred 5'-3' polarity of RNA bound to RRM1 in the human protein is indicated. (G) Predicted aligned error maps and domain schematics of yeast Rex1 and the fly and human Rexo5 proteins.

(RRM) domains within their C-terminal regions that are not seen in Rexo5 from *D. melanogaster* or yeast Rex1 [14,15] (Fig 1D–F). Comparison of high confidence AF models for Rexo5 proteins from *D. melanogaster* and humans (mean pLDDT scores of 84 and 80 for the residues shown in Fig 1E and F, respectively) revealed the same DEDDh/AlkP dual domain structure seen in yeast Rex1. Furthermore, structure comparisons using the Dali server revealed that the modelled AlkP domains of the fly and human proteins show high Z-scores and rmsd values (12.2 and 3.1Å for AF-Q9VRX7 and 13.0 and 2.2Å for AF-Q96IC2, respectively) with the experimentally determined structure of iPGM from *Bacillus stearothermophilus* [24] (S1 Table). Notably, the degree of packing between the DEDDh and AlkP domains is less in the model of the fly protein than the yeast protein, and this is further decreased in the human protein (Fig 1G). This leads to variation in the mutual orientation of these domains in the models.

## Insertions within the AlkP domain of Rex1-related proteins

Non-conserved sequences within the AlkP superfamily map as insertions on the surface of the common α/β fold [26]. The central "cap" domain of PPMs and iPGMs is inserted between β2 and β3 of the AlkP domain (Jedrzejas et al. 2000; Panosian et al. 2011). Strikingly, the DEDDh domain within the dual domain structure of Rex1-related proteins is also inserted between β2 and β3 (Fig 1B). This common insertion point is consistent with observations that the interface between the AlkP domain and the cap or DEDDh domain may have functional significance. Both the AlkP and cap domains contribute to the catalytic centres of PPMs and iPGMs [24,25]. The discontinuous domain of Rex1-related proteins is not predicted to have alkaline phosphatase activity, since it lacks the conserved serine or threonine residue that is involved in phosphotransfer (Galperin & Jedrzejas, 2001). We predict, however, that the AlkP domain of Rex1 directly contributes to substrate binding on the basis of our earlier *in vivo* crosslinking experiments and structural threading studies [18].

The relative size and complexity of the loop insertions varies across Rex1-related proteins. Loop 3 (connecting β6 and β7) is extended in yeast and plant proteins but contracted to a short loop in Rexo5 from *D. melanogaster* and humans (Fig 1 B,E,F). Loop 2 (at the C-terminal end of β5) is also contracted in the fly protein but expanded within the vertebrate proteins to include the RRMs. We observed that Rex1-related proteins from trypanosomes are predicted to have an additional extended loop between strands 1 and 2 (see S2 Fig).

The predicted AF structures of the two RRMs within the human Rexo5 protein and homologues from mouse (*Mus musculus*), frog (*Xenopus laevis*), chick (*Gallus gallus*) and fish (*Danionella translucida*) were compared with protein structures within the PDB using the Dali server (S3 Table). The Rexo5 RRM model structures from each species tested have very high similarities with the twin RRMs that constitute the *half pint* domain (Dali Z scores of 26.5–17.2 and rmsd values of 2.5–1.3 Å) found within PUF60 and FIR proteins [30] (Fig 2). The Z scores observed for structures of other RRM-containing proteins were notably lower (shown for the human protein in Fig 2, see S3 Table) and matched a single RRM domain. Furthermore, models of the human and mouse Rexo5 proteins were the most similar structures identified in searches comparing the PUF60 RRM structure (5KWQ) against species-specific AF databases (S3 Table). Poly(U)-binding splicing factor 60 (PUF60) and FUSE-binding protein (FBP)-interacting repressor (FIR) are protein isoforms that

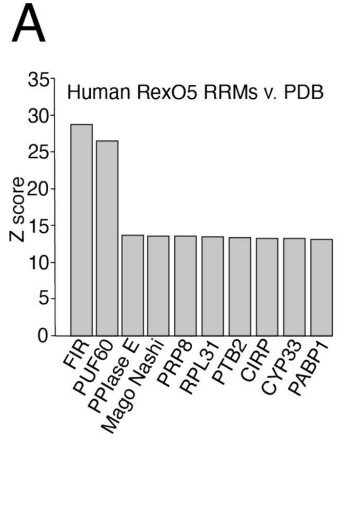

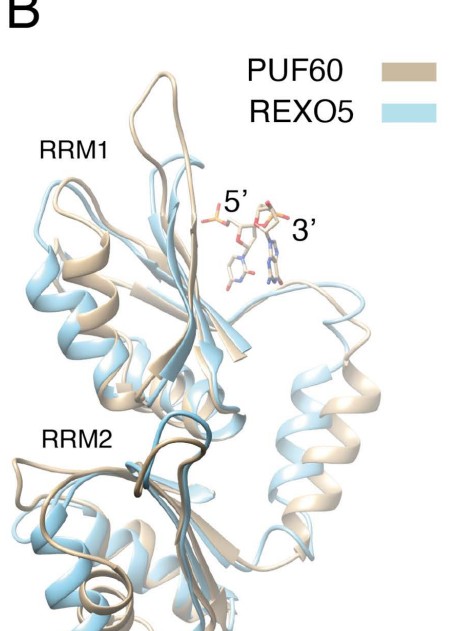

| | AF model | PUF60 Z | rmsd |
|---|---|---|---|
| Rexo5 (human) | Q96IC2 | 26.5 | 1.3 |
| Rexo5 (mouse) | D3YW29 | 17.2 | 2.3 |
| Rexo5 (*frog*) | A0A1L8EQK4 | 17.5 | 2.4 |
| Rexo5 (*chick*) | Q5ZMH0 | 18.9 | 2.5 |
| Rexo5 (*fish*) | A0A553Q8D6 | 17.9 | 2.4 |

**Fig 2. The RRM domains of vertebrate Rexo5 proteins are related to PUF60/FIR.** (A) A plot of Dali Z scores for the twin RRMs within the human Rexo5 protein AF model and the most closely related protein structures within the PDB. (B) Structural overlay of the RRM domains in the AF model of human Rexo5 (AF-Q96IC2) and the resolved structure of PUF60 with bound dinucleotide (PDB ID 5KW1) [33]. Z-scores and rmsd values for PUF60 structure overlays with AF models of vertebrate REXO5 proteins are shown.

share the twin RRMs. This suggests that vertebrate Rexo5 and PUF60/FIR proteins have a degree of structural similarity that is not shared with other proteins containing adjacent RRMs such as polypyrimidine tract binding protein (PTB) and poly(A) binding protein 1 (PABP1) [31,32].

The N-terminal RRM of PUF60/FIR (labelled RRM1 in Fig 2) is packed against the C-terminal RRM, obstructing its RNA binding surface [33,34]. This packed alignment is dependent upon a long α-helix connecting the two RRMs [34] that is also predicted to be present within Rexo5 structures (Figs 1, 2). The low predicted aligned error within the AF model of human Rexo5 for the C-terminal region of the protein (Fig 1G) is consistent with a tight packing between the twin RRM domains and a close alignment of the RRMs with the AlkP domain. The orientation of bound nucleic acid on consensus RRMs is conserved [35]. Despite the low degree of packing between the DEDDh and AlkP domains, the AF model of human Rexo5 is consistent with bound RNA passing across the surface of the N-terminal RRM towards the active site of the DEDDh domain in a 5'-3' direction (Figs 1F, 2B).

## Loops at the domain interface of Rex1 are required for 5S rRNA and tRNA processing

A striking feature of the yeast Rex1 model is the three prominent loops on the surface of the AlkP domain that are projected towards the catalytic DEDDh domain (Fig 1A,B). Comparison of AF models of Rex1 proteins from ascomycota (*S. cerevisiae*, *E. nidulans*), basidiomycota (*S. pombe*, *C.neoformans*) and mucuromycota (*R. irregularis*) (Fig 3A) reveal that these features are conserved across fungal species. Loop 3 forms an extended helical arch structure and is clearly identifiable in the AF models within structural similarity clusters of Rex1-related proteins from fungi, amoebozoa, diatoms, plants and green algae (S1 Table).

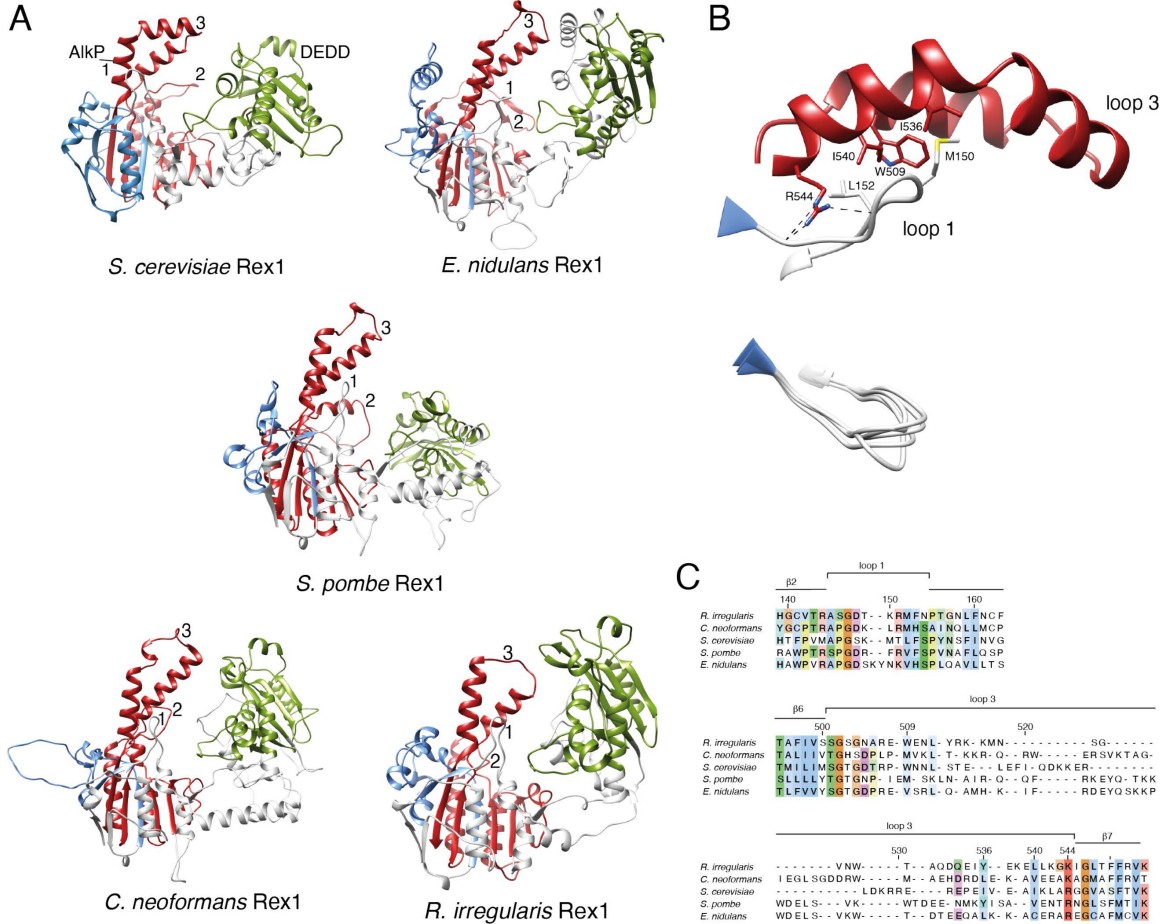

**Fig 3. Surface Loops on the AlkP domain of Rex1 are conserved across fungi.** (A) Comparison of the AF model of Rex1 from *S. cerevisiae* (AF-P53331) (Fig 1A) with homologous proteins from *E. nidulans* (AF-Q5AVW4), *S. pombe* (AF-O94443), *C. neoformans* (AF-A0A225XBF6) and *R. irregularis* (AF-A0A2I1EHK1). The AlkP and DEDD domains are coloured as in Fig 1. Unstructured N- and C-terminal regions of the protein models are omitted for clarity. The positions of loops 1, 2 and 3 are indicated. (B) Expanded view of the interaction between loop 1 and loop 3 (residues 144-155 and 504-544, respectively) in Rex1 from *S. cerevisiae*. The positions of key residues are indicated. Dashed lines indicate hydrogen bonding between loops 1 and 3. A structural overlay of loop 1 sequences from the five fungal proteins is included. (C) AF model structure-based sequence alignment of loops 1 and 3. Residues are coloured by conservation.

Loop 1 and loop 3 residues within the AF model of yeast Rex1 have very high confidence mean pLDDT scores (93 for both polypeptide sequences) and are packed closely together to form an integrated structural module (Fig 3B). Alignment of loop 3 sequences with high confidence scores in the AF models (minimum mean pLDDT score of 78 for the sequences shown in Fig 3) revealed residues conserved across fungi that are predicted to either make main chain interactions with loop 1 (R544) or to be closely packed against it (W509, I536, I540) (residue numbers given for the *S. cerevisiae* protein, see Fig 3C). Residues within loop 1 are more highly conserved than residues within loop 3 across fungal species and are modelled to adopt a common structure (Fig 3B). This suggests that loops 1 and 3 in Rex1 form a structural module that is broadly conserved across fungi.

We replaced loops 1–3 within Rex1 from *S. cerevisiae* with short linker peptides and assayed the ability of the alleles to complement the synthetic lethal growth phenotype of a *rex1Δ rrp47Δ* double mutant and the 5S rRNA and tRNA processing defects observed in a *rex1Δ* mutant. Wild-type and mutant *rex1* alleles proteins were cloned into a plasmid bearing the

*HIS3* marker and the proteins were expressed as N-terminal fusions to assess effects of the mutations on protein expression levels.

The wild-type *REX1* allele supported growth of the *rex1Δ rrp47Δ* double mutant in plasmid shuffle assays, whereas there was no growth observed for the control vector transformant on medium containing 5'-fluoroorotic acid (FOA) (Fig 4A). Strains expressing either the loop1Δ, loop2Δ or loop3Δ mutant were viable on FOA medium. Three independent FOA-resistant *rex1 rrp47Δ* isolates expressing each of the loop mutants showed comparable growth to isogenic isolates expressing the *REX1* wild-type allele on YPD medium (Fig 4A). We also carried out random spore analyses of the meiotic progeny derived from a heterozygous *rex1Δ::KANMX4/REX1 rrp47Δ::HYGMX4/RRP47* diploid strain. Hygromycin- and G418-resistant progeny were recovered from transformants encoding the loop1Δ, loop 2Δ and loop 3Δ mutants but not from the nontransformed parental strain (Fig 4B, Table 1). All meiotic progeny scored as being *rex1Δ rrp47Δ* double mutants in Table 1 showed clear growth on medium containing hygromycin, G418 and both antibiotics in combination, were viable on medium lacking histidine and were shown to express a Rex1 fusion protein by western analysis. A single isolate from the vector-transformed strain exhibited growth on medium containing both hygromycin and G418. However, RNA analyses revealed that this isolate accumulated the extended form of 5S rRNA that is characteristic of *rex1Δ* mutants but did not accumulate the extended form of 5.8S rRNA that is observed in *rrp47Δ* mutants [13]. We therefore conclude that this strain is a *rex1Δ* single mutant that has acquired resistance to hygromycin. Our data are consistent with the reported synthetic lethality of the *rex1Δ rrp47Δ* double mutant and demonstrate that each *rex1* loop deletion mutant complements this phenotype.

The relative electrophoretic mobilities of the wild-type and mutant proteins upon western analyses were consistent with the predicted molecular weights of the fusion proteins (wild-type, 77.6kDa; loop 1Δ, 77kDa; loop 2Δ, 76.5kDa; loop 3Δ, 73.2kDa). The expression levels of the three mutants were similar and ∼ 60–80% of the expression level of the wild-type protein. This reduction is not predicted to be limiting for Rex1 function, since expression of plasmid-borne fusion proteins is more than three times higher than the expression level of a fully functional, chromosomally encoded fusion protein [18].

Yeast 5S rRNA is extended at the 3' end by 3–4 nucleotides in the *rex1Δ* mutant, compared to a wild-type strain [3,4]. Northern blot hybridisation analyses of total cellular RNA revealed that the 5S rRNA processing defect seen in the *rex1Δ* mutant was complemented upon expression of the wild-type *REX1* allele or the loop 2Δ mutant but not in strains expressing the loop 1Δ or loop 3Δ mutant (Fig 4D).

Strains lacking Rex1 are also defective in the 3' end maturation of tRNA$^{Arg}_{UCU}$ that is processed from dicistronic tRNA$^{Arg}_{UCU}$-tRNA$^{Asp}_{GUC}$ transcripts [3,36]. Processing of the ∼ 170 nucleotide long, primary transcript involves initial RNase P cleavage at the 5' end of tRNA$^{Asp}_{GUC}$ to generate a ∼ 90 nucleotide long, 5'- and 3' extended tRNA$^{Arg}_{UCU}$ intermediate that is subsequently processed at its 5' end by RNase P and at its 3' end by Rex1 [36,37]. Hybridisation using a probe complementary to both the 3' end of tRNA$^{Arg}_{UCU}$ and the downstream trailer sequence detected bands in RNA from the wild-type strain corresponding in length to the primary transcript, tRNA$^{Arg}_{UCU}$ processing intermediates and the mature tRNA (Fig 4D). We also analysed RNA from the *pop1–1* temperature sensitive mutant to provide additional insight into the nature of the detected bands. The *POP1* gene encodes a protein subunit of both RNase P and RNase MRP and the *pop1–1* temperature sensitive allele shows a nonconditional defect in RNase P activity that is exacerbated upon growth at the nonpermissive temperature [38]. As predicted for moderate inhibition of RNase P activity during growth at the permissive temperature, the *pop1–1* mutant grown at 23°C accumulated the primary transcript and the ∼ 90 nucleotide long tRNA$^{Arg}_{UCU}$ processing intermediates, as well as a slightly shorter fragment not observed in the wild-type strain that results from 3' end processing in the absence of RNase P cleavage [37]. Consistent with decreased RNase P activity upon depletion of Pop1, growth at the nonpermissive temperature led to further accumulation of the 3' processed dicistronic transcript and depletion the tRNA$^{Arg}_{UCU}$ processing intermediates. As previously observed [3], the *rex1Δ* mutant accumulated 3' extended forms of both the mature tRNA$^{Arg}_{UCU}$ and the processing intermediates. Similar to the pattern observed for 5S rRNA processing, the tRNA$^{Arg}_{UCU}$ processing defects observed in the *rex1Δ* mutant were complemented upon expression of the wild-type *REX1* allele or the loop 2Δ mutant but not in strains expressing the loop 1Δ or loop 3Δ mutant (Fig 4D).

 

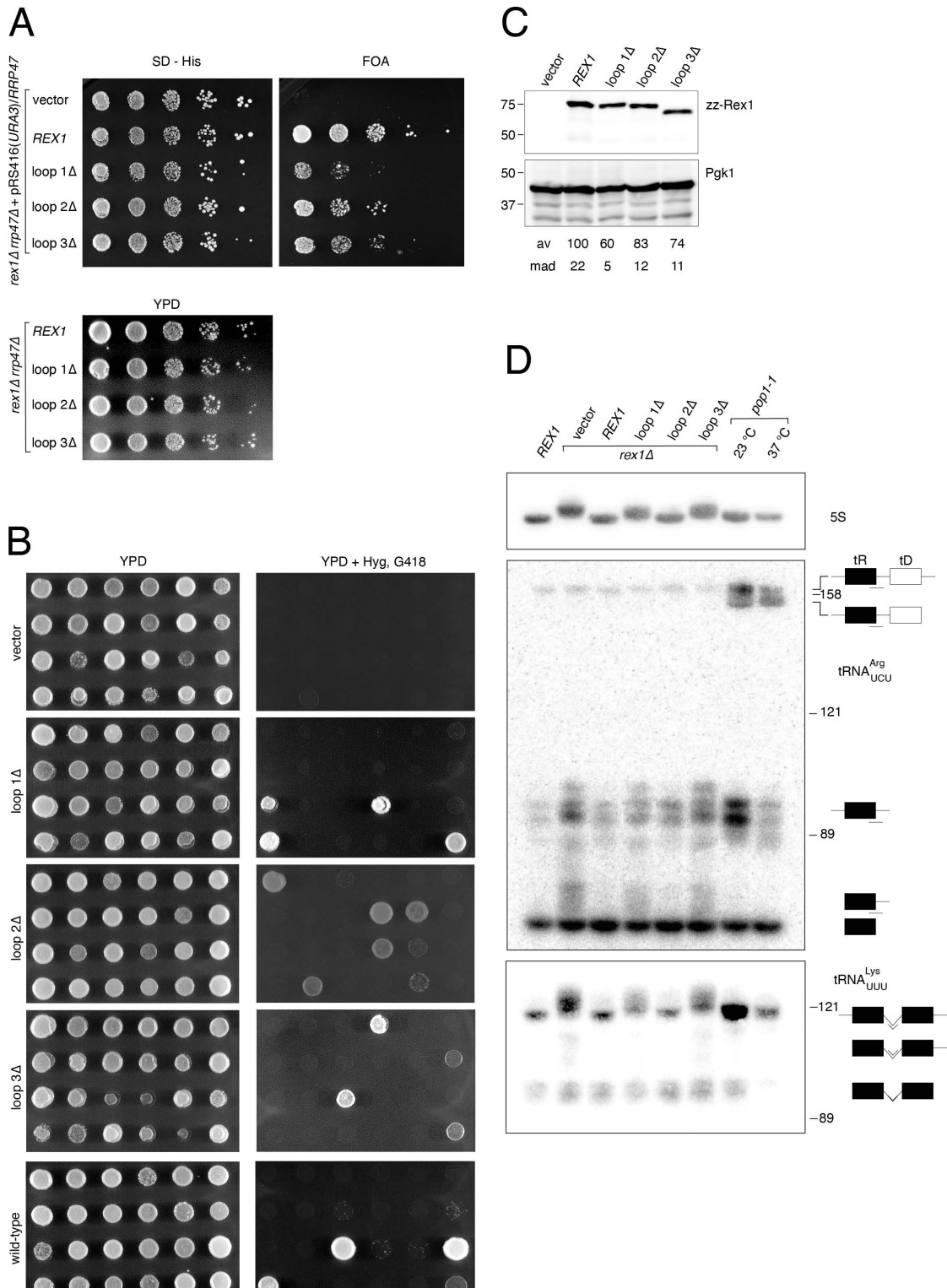

**Fig 4. Loops within the AlkP domain of Rex1 are required for RNA processing *in vivo*.** (A) Spot growth assays of *rex1Δ rrp47Δ* double mutants. Upper panels, growth of a *rex1Δ rrp47Δ* strain harbouring a plasmid expressing the *RRP47* and *URA3* genes after transformation with a *HIS3* marker plasmid encoding *rex1* alleles. Growth on selective medium lacking histidine (SD - His) or containing FOA to purge the *URA3* marker plasmid was

recorded after 3 days. Lower panel, growth of FOA-resistant isolates on YPD medium. Plates were photographed after 2 days incubation. Assays were performed on three independent isolates. (B) Spot growth assays of randomly selected meiotic progeny from a *rex1Δ::KANMX4/REX1 rrp47Δ::HYGMX4/ RRP47* diploid strain harbouring plasmids expressing the wild-type *REX1* gene, the loop deletion mutants or the control vector. The panels show growth of haploid progeny on nonselective YPD medium and on medium containing hygromycin and G418. Plates were photographed after incubation for 1-2 days. (C) Western blot analyses of *rex1* mutant protein expression levels. Wild-type and mutant *rex1* proteins were expressed in a *rex1Δ* single mutant grown at 30°C. Electrophoretic mobilities of molecular weight markers (size in kDa) are indicated on the left-hand side. of each blot. Rex1 expression levels were normalised to Pgk1. Values for the mean average (av) and mean absolute deviation (mad) of three independent replicates are given. (D) Northern blot hybridisation analyses of total cellular RNA from a wild-type strain, *rex1* mutants and the *pop1-1* temperature-sensitive mutant. The north-ern blot was hybridised reiteratively with oligonucleotides complementary to the acceptor stem/3' trailer sequence boundary of unprocessed tRNA$^{Arg}_{UCU}$, the tRNA$^{Lys}_{UUU}$ intron and 5S rRNA. Detected tRNA species are indicated schematically on the right. Lines under the schematics indicate the positions of sequences complementary to the oligonucleotide probes used for each panel. The electrophoretic mobilities of 5S rRNA (121 nucleotides), 5.8S$_S$ (158 nucleotides) and U24 (89 nucleotides) are indicated as size markers.

**Table 1. Random spore analysis of meiotic progeny from heterozygous *rex1Δ/REX1 rrp47Δ/RRP47* strains harbouring *rex1* loop mutants.**

|  | nontransformed | + vector | + wild-type | + loop1Δ | + loop 2Δ | + loop 3Δ |
|---|---|---|---|---|---|---|
| number of progeny scored | 82 | 93 | 90 | 87 | 91 | 86 |
| wild-type | 50 | 31 | 28 | 31 | 29 | 28 |
| *rex1Δ* | 19 | 32 | 31 | 21 | 21 | 24 |
| *rrp47Δ* | 13 | 30 | 26 | 32 | 30 | 28 |
| *rex1Δ rrp47Δ* | 0 | 0 | 5 | 3 | 11 | 6 |

The *rex1Δ* mutant is also defective in the 3' maturation of tRNA$^{Lys}_{UUU}$ [6,9]. Intron containing tRNAs in yeast are spliced in the cytoplasm after nuclear 5'- and 3' processing events. Yeast tRNA$^{Lys}_{UUU}$ primary transcripts are ~120 nucleotides long and contain a 23 nucleotide long intron [39]. A probe specific to the intronic sequence of tRNA$^{Lys}_{UUU}$ detected two major species in wild-type cells with lengths consistent with that of the primary transcript and the 5'- and 3' end-processed but unspliced intermediates [6,9]. Compared to the wild-type strain, the *pop1–1* mutant accumulated the primary tRNA$^{Lys}_{UUU}$ primary transcript during growth at the permissive temperature and was depleted of the end-processed intermediates upon transfer to the nonpermissive temperature. As previously reported [6,9], the *rex1Δ* mutant accumulated extended forms of the tRNA$^{Lys}_{UUU}$ primary transcript and species that migrated between the primary transcript and the end-processed intermediate (Fig 4D). The accumulation of species longer than those observed in the wild-type strain reflects both a block in processing in the *rex1Δ* mutant and the oligoadenylation of the unprocessed substrates, as has been pre-viously reported for both 5S rRNA and tRNAs [4,7]. Comparison of the northern blot hybridisation patterns revealed that defects in tRNA$^{Lys}_{UUU}$ processing observed in the *rex1Δ* mutant were complemented by the wild-type *REX1* allele and the loop 2Δ mutant but not upon expression of the loop1Δ or loop3Δ mutant (Fig 4D).

Taken together, these data show that loop 1 and loop 3 are required for Rex1-mediated 5S rRNA and tRNA processing but not for growth in a Rex1-dependent strain. These observations are not inconsistent, as the defect(s) in RNA process-ing and/or turnover underlying the synthetic lethal growth phenotype of *rex1Δ rrp47Δ* strains is not related to 5S rRNA or tRNA processing [6,11].

### Rex3 and REXO1 proteins share a common domain architecture

In contrast to Rex1, the DEDDh domain of Rex3 is found at the C-terminus of the protein. Structural overlay of AF models for yeast Rex1 and Rex3 (mean pLDDT score of 88 for the entire length of the protein) suggested two prominent features of Rex3 that are not found in Rex1-related proteins (Fig 5A).

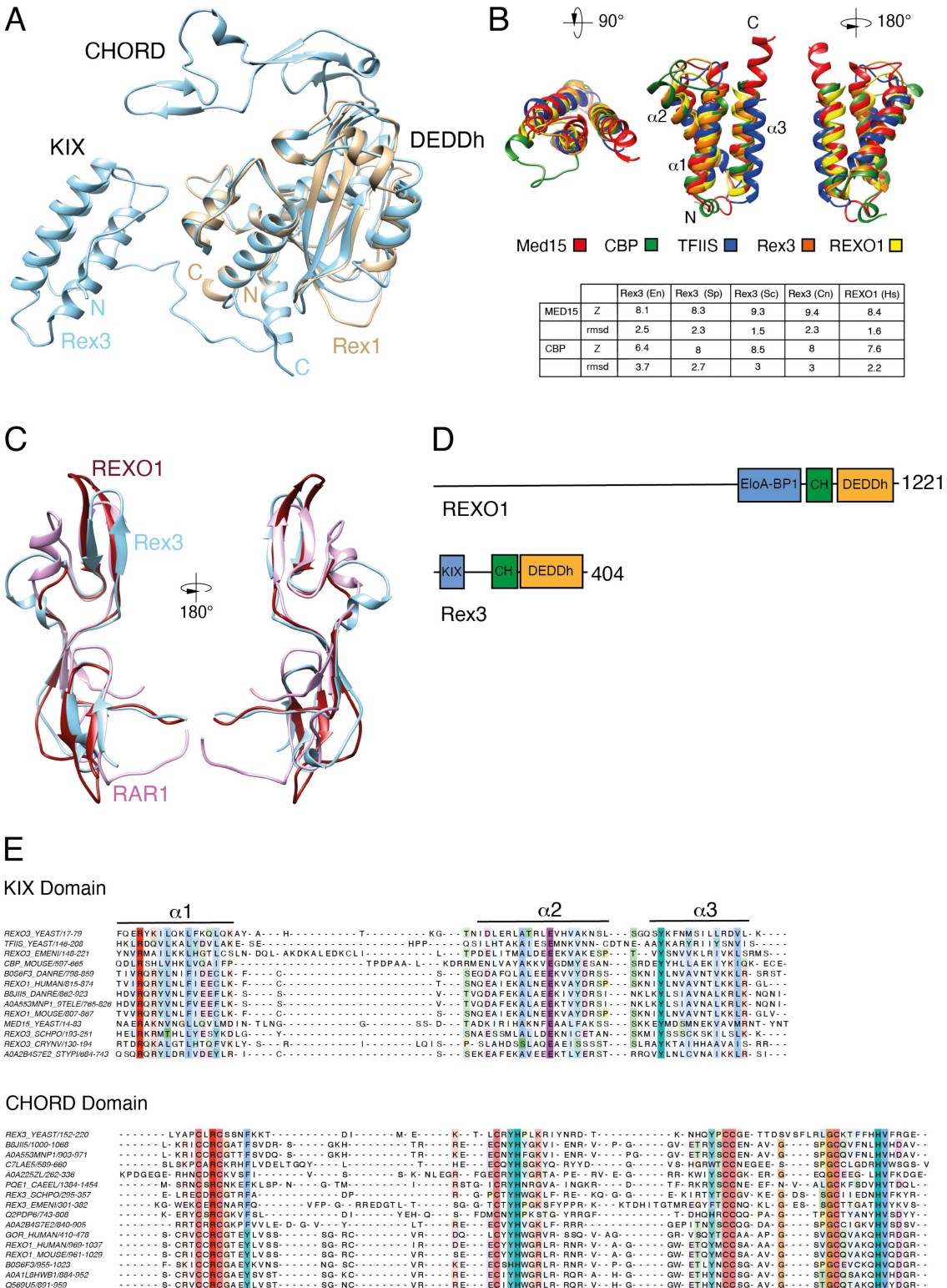

**Fig 5. Rex3 related proteins contain KIX and CHORD domains.** (A) Structure overlay of AF models for full-length Rex3 (blue) and the DEDDh domain of Rex1 (brown) from *S. cerevisiae*. Protein domains and the N'- and C'-termini are labelled. (B) Overlay ribbon diagrams of the triple helical domains within yeast Rex3 and human REXO1 proteins and the KIX domains of yeast Med15, mouse CBP and yeast TFIIS. Z-scores and rmsd values

for the correlation between Rex3/REXO1 protein structural models and the structures of the KIX domains of Med15 and CBP are given. (C) Structural overlays of the AF models for the CHORD domain of yeast Rex3 (blue) and human REXO1 (red) with the crystal structure of the CHORD domain of RAR1 (purple) [40]. (D) Domain structure of human REXO1 and yeast Rex3 proteins. (E) Structure-based multiple sequence alignments of the KIX and CHORD domains of Rex3 and REOX1 proteins. Residues are coloured by conservation. The KIX domain alignment includes TFIIS, Med15 and CBP.

Firstly, the N-terminal region of yeast Rex3 is predicted to form a compact triple helix. Structure comparisons with PDB entries using the Dali server revealed a high degree of similarity between this predicted domain and the KIX domain of the mediator subunit Med15/Gal11 of *Candida glabrata* and with mouse CREB-binding protein (CBP) (Fig 5B, S4 Table). Reverse structural homology searches identified similarity between the KIX domain of Med15 and Rex3 proteins within *S. cerevisiae* and *S. pombe* (S4 Table). The helices of the KIX domain are numbered 1–3 in Fig 5, from the N- to C-terminus. The structural fold of characterised KIX domains is dependent upon a conserved cation-π interaction between an arginine at the base of helix 1 and a tyrosine at the base of helix 3 [21]. These residues are conserved within the helical domains seen in Rex3-related proteins (Fig 5E). Furthermore, a glutamate residue within helix 2 that is modelled within the AF structures to interact with both the arginine and tyrosine residues is also conserved.

The KIX domain within Med15 and CBP enables kinase-responsive protein interactions with disordered regions of transcription factors [21]. Other proteins that contain KIX domains include the DNA helicase RECQL5 and the transcription elongation factor TFIIS, both of which have been shown to interact directly with RNA polymerase II (RNAP II) [41]. Interaction between RECQL5 and RNAP II leads to repression of transcription elongation. This is due, in part, to a competition between the KIX domains within RECQL5 and TFIIS for binding to the jaw domain of the Rpb1 subunit of RNAP II [41]. TFIIS interacts with RNAP II through its central KIX domain (domain II) and induces the endonucleolytic cleavage of backtracked transcripts within stalled complexes through its C-terminal catalytic domain, enabling reactivation of transcription [42]. By analogy with TFIIS, one possibility is that the KIX domain of Rex3 may allow its recruitment to Rpb1 and promote coupling of transcription termination with subsequent 3' processing or degradation of the transcript. Notably, RNAP II is enriched with Rex3 in yeast mutants that are depleted of the transcription elongation factor Spt6 [43] while the human protein REXO1 is highly enriched in promoter-proximal, paused RNAP II checkpoint complexes [44].

In addition to the KIX domain, yeast Rex3 harbours a ~ 70 residue long, extended domain comprising three pairs of short antiparallel β-sheets that is positioned immediately upstream of the catalytic DEDDh domain. Structural homology searches within PDB revealed a common fold between this region and the cysteine- and histidine-rich domains (CHORD) found in CHORD-containing protein 1 and the Hsp90 cochaperone Rar1 [40] (Dali Z-score of 4, rmsd of 3.3 Å) (S4 Table). CHORD domains consist of two $C_3H$ motifs, containing three cysteines and a histidine residue, that each bind a $Zn^{2+}$ ion [19,20]. They mediate specific protein-protein interactions, notably between Hsp90 and the cochaperone protein Sgt1 [19,40,45]. The cysteine and histidine residues are conserved throughout the CHORD domain in Rex3-related proteins, as is their relative spacing (Fig 5E), strongly suggesting that Rex3 is a zinc-binding protein. The CHORD domains of Rex3-related proteins are unusual inasmuch as they are single domains; CHORD domains are typically observed as tandem domains. CHORD domains are found in proteins from metazoans, plants and *Toxoplasma* but, notably, were previously reported as being absent from *S. cerevisiae* [19]. Reverse structural homology searches suggest Rex3 is the only CHORD domain containing protein in *S. cerevisiae* and *S. pombe* (S4 Table).

The human DEDDh enzymes REXO1 and GOR were identified through structural comparision of the KIX and CHORD domains of yeast Rex3, the KIX domain of Med15 and the CHORD domain of RAR1 (S4 Table). The AF model of REXO1 overlaps well with KIX domains within Med15 and CBP (Fig 5B), and with the CHORD domain of RAR1 (Fig 5C). Furthermore, the conserved arginine, glutamate and tyrosine residues within the KIX domain and the cysteine and histidine residues of the CHORD domains are found in REXO1 (Fig 5E). Thus, the human enzymes REXO1 and GOR have extensive structural homology with yeast Rex3 while their homology to Rex1 and Rexo5 is limited to the DEDDh catalytic domain.

REXO1 was initially characterised as Elongin A binding protein 1 (EloA-BP1), a protein that interacts with the structurally related N-terminal domains of transcription factors Elongin A and TFIIS [17]. The Elongin A binding-protein 1 (EloA-BP1) domain (PF15870) annotated in Pfam and the RNA exonuclease 1 homologue-like domain annotated in Inter-Pro (IPR031736) span the helix bundle of the KIX domain and extend up to the CHORD domain, including a short helical stretch (residues 119–142 in yeast Rex3) that is packed against the DEDDh domain and conserved between Rex1 and Rex3 proteins. The EloA-BP1 domain is not only found in vertebrate and invertebrate REXO1 proteins but is also annotated in InterPro for Rex3 proteins from the yeasts *S. pombe*, *Kluyveromyces lactis*, *Yarrowia lipolytica* and *Cryptococcus neoformans*. This further supports the structural homology seen across yeast Rex3 and REXO1 proteins.

## The N-terminal region of Rex3 is required for 3' processing of RNase MRP RNA

Little is known about the cellular substrates of REXO1 and Rex3 proteins. Yeast Rex3 is required for the accurate 3' end processing of the RNA component of RNase MRP and functions redundantly with Rex1 and Rex2 in the processing of RNase P RNA, U5 snRNA and the autoregulated degradation of *RTR1* mRNA [3,10]. Studies in mammalian cells link the function of elongin A to transcription of enhancer RNAs and rRNA synthesis [46]. To address the requirement for the N-terminal region of Rex3 for its function *in vivo*, we expressed full-length and N-terminal truncated versions of Rex3 as fusion proteins in yeast and assayed the profiles of RNase MRP RNAs and the expression levels of the fusion proteins. Two *rex3* deletion constructs were generated, one which lacks a region spanning the KIX domain (Δ1–106) and one which expresses essentially only the catalytic domain (Δ1–218) (Fig 6A).

Yeast RNase MRP RNA is 340 nucleotides long, with a single nucleotide heterogeneity at its 3' end [47]. The transcript is ~7 nucleotides longer in the *rex3Δ* strain [3]. Northern blot hybridisation analyses confirmed that RNase MRP RNA is longer in the *rex3Δ* mutant than an isogenic wild-type strain and showed that expression of the full length Rex3

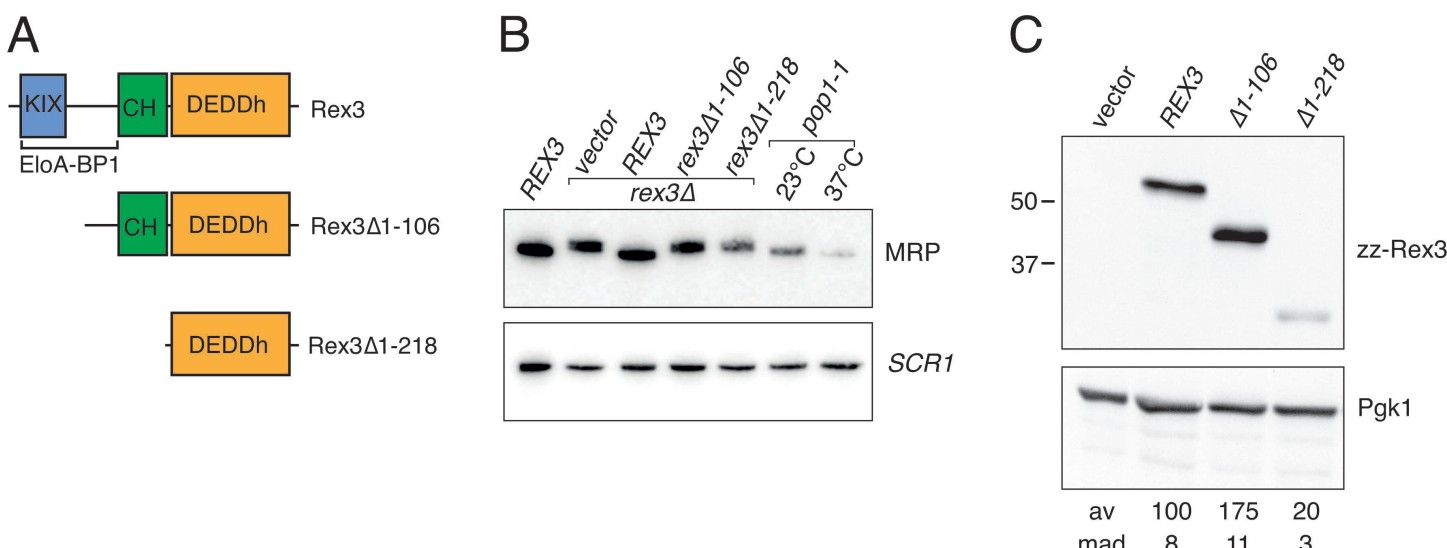

**Fig 6. The N-terminal region of Rex3 is required for processing of RNase MRP RNA.** (A) Schematic of the domain architecture of yeast Rex3 and the deletion mutants analysed in this study. (B) Northern blot analyses of RNA from a *REX3* wild-type strain, isogenic *rex3Δ* mutant strains harbouring the cloning vector or expressing either full length, wild-type Rex3 or the deletion mutants, and the *pop1-1* mutant during growth at 23°C and 37°C. Total cellular RNA was resolved through 6% polyacrylamide gels, transferred to Hybond N⁺ membranes and hybridised with a probe specific to the RNase MRP RNA. An *SCR1* probe was used as a loading control. (C) Western analyses of whole cell lysates from strains expressing full length Rex3 or the N-terminal deletion mutants. Rex3 expression levels were normalised to Pgk1 and are expressed as a percentage of the wild-type protein. Mean average (av) and mean absolute deviation (mad) values of three independent replicates are given.

fusion protein fully complemented this processing defect (Fig 6B). In contrast, the unprocessed form of RNase MRP RNA accumulated upon expression of either of the *rex3* deletion mutants (Fig 6B). Deletion of the N-terminal 106 residues of Rex3 caused a marked increase in its steady state expression levels, whereas expression of the catalytic domain alone (*rex3Δ1–218*) led to a clear reduction (Fig 6C). The *pop1–1* mutant did not show a defect in RNase MRP RNA processing but the RNA was depleted upon incubation at the nonpermissive temperature for this mutant (Fig 6B), as reported previously [38].

We conclude that the N-terminal region of yeast Rex3 (residues 1–106) spanning the KIX domain is required for its role in RNase MRP RNA processing. Loss of RNase MRP processing in this mutant is consistent with a role for the KIX domain in the recruitment of Rex3 to transcriptional complexes described above. The mature RNase MRP RNA sequence ends with a single-stranded stretch of three to four C residues [47,48]. This sequence is immediately followed in the genomic DNA by a T tract that functions as an effective termination signal for both RNA polymerases II and III [49]. This suggests that Rex3 removes the short U tail at the 3' end of the primary transcript and is strongly inhibited upon encountering consecutive cytidine residues. Other well characterised DEDD family exoribonucleases have been shown to be inhibited by short C tracts [50,51], suggesting this may be a widespread characteristic of the activities of these enzymes.

Why an increase in steady state levels of Rex3 is observed in the *rex3Δ1–106* mutant is not clear but a plausible explanation is that Rex3 autoregulates its expression in a manner analogous to its role in the regulation of the *RTR1* mRNA [10]. The larger Δ1–218 deletion caused a marked decrease in Rex3 expression levels (Fig 6C). Residues 107–218 of Rex3 include a short region (residues 119–142) that is packed against the DEDDh domain in the AF model, in addition to the CHORD domain (residues 152–220). One explanation for the low expression level of the Δ1–218 mutant is that Rex3 stability is dependent upon an intramolecular interaction between residues 119–142 and the catalytic domain. Alternatively, Rex3 stability may be affected by loss of the CHORD domain.

### Rex1 homologues are ubiquitous in eukaryotes while Rex3 proteins are restricted to fungi and animals

Sequence alignments can show some variability when analysing divergent protein domains. However, analyses of 387 selected DEDDh domain proteins from a broad spectrum of eukaryotic organisms revealed a distinct clade that comprises the annotated REXO1, Sdn5, Rex1, Rexo5 and Rex3 proteins (S5 Fig). A phylogenetic analysis of this clade, together with a multiple sequence alignment of their Exo I, II and III motifs, is shown in Fig 7. The presence of the AlkP, RRM, KIX and CHORD domains within the AF models of each protein and their pLDDT scores are indicated on the right of the alignment. An analysis of the confidence scores for the domains within each model and links to the structures are given in S6 Table. A multiple sequence alignment of the KIX and CHORD domains of Rex3 proteins is shown in S7 Fig.

The Rex1/Rex3 family can be divided into two distinct clades on the basis of both sequence and structure criteria. Rex1-related proteins are found in all major eukaryotic lineages, including fungi, amoebae, metazoans, plants, green algae and in the protists *Trypanosoma brucei* (Discoba clade) and *Tetrahymena thermophila* (SAR clade). This suggests that the ancestral *REX1* gene arose early during eukaryotic evolution. No Rex1 homologue was found for *Danio rerio* but a homologue was identifiable in the closely related cyprinid fish, *D. translucida.* Similarly, we did not identify a Rex1 homologue in the genomes of the green alga *Chlamydomonas reinhardtii* or the spikemoss *Selaginella moellendorffii* but homologous proteins were identified in the algae *Auxenchlorella protot*hecoides*, *Klebsormidium nitens* and *Chlorella sorokiniana*, as well as the stonewort *Chara braunii* and common liverwort, *Marchantia polymorpha*. *M. polymorpha* encodes two Rex1-related proteins, one that has a predicted structure typical of the Rex1 family and one that is essentially just the DEDDh domain (A0A2R6WS43 and A0A2R6WS50, respectively). Of the thirty one AlkP domain structural models analysed, two (I7LTR4 from *T. thermophilus* and A0A1D1ZUZ2 from *A. prototheocoides*) were of low confidence (pLDDT scores below 70) and five were partial models that lacked peripheral strands at either the "ß2 end" (A0A3M7KUZ0 from *A. prototheocoides* and A0A388LS10 from *C. braunii*) or the "ß4 end" (K8EFH5 from *B. prasinos*, I7LTR4 from *T. thermophila* and Q5ZMH0 from chick) (Fig. 7 and S6 Table). No Rex1-related protein model contained either a KIX or CHORD domain.

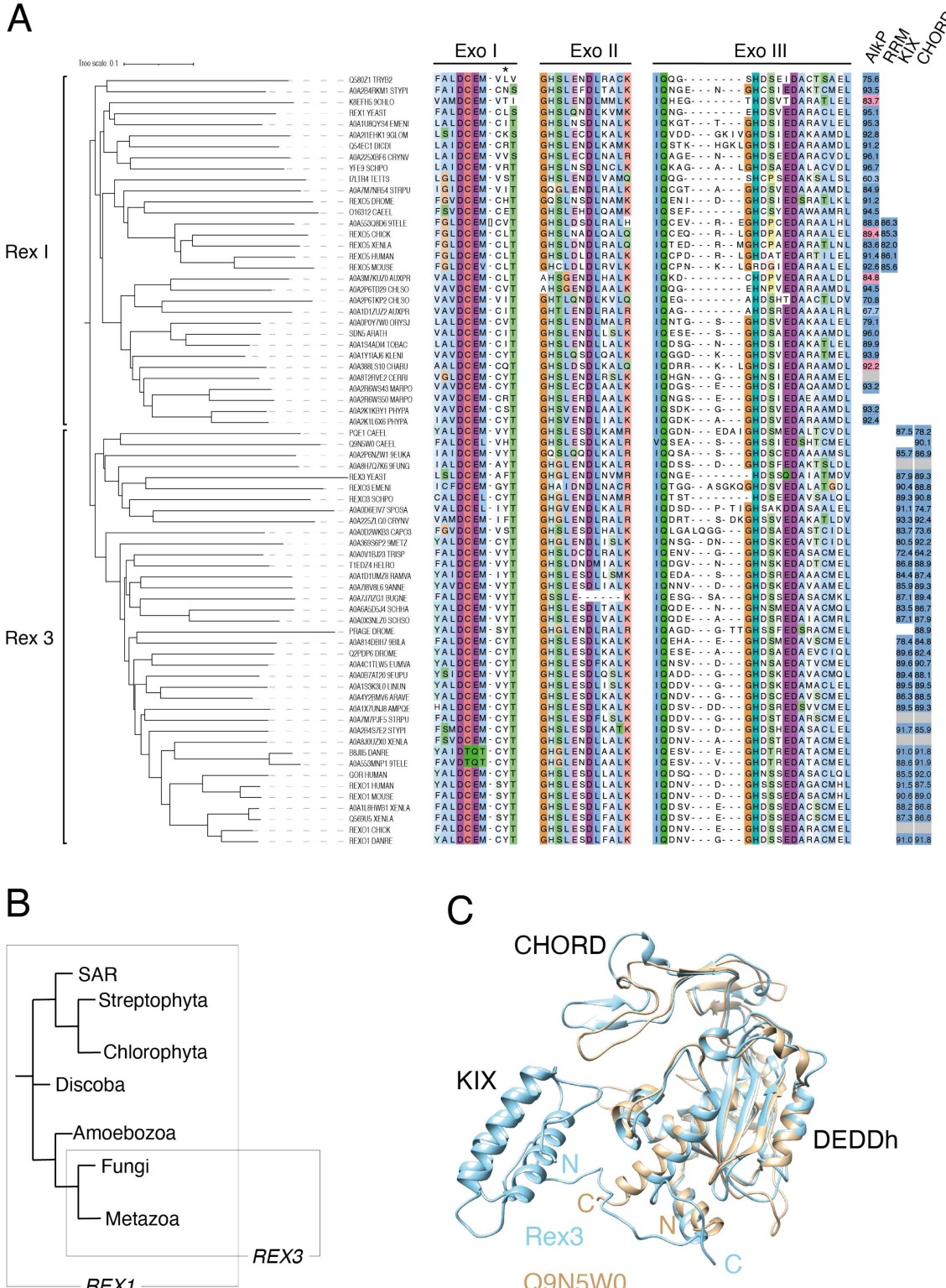

**Fig 7. Phylogenetic analyses of Rex1 and Rex3 proteins.** (A) Phylogenetic tree and multiple sequence alignment of Rex1 and Rex3 proteins. Exo I, II and III motif sequences are shown. The asterisk indicates a conserved Y/F residue in Rex3 proteins. The presence of AlkP, RRM, KIX and CHORD domains within AF structures are indicated to the right (dark blue indicates high confidence pLDDT scores above 70; light blue indicates low confidence

pLDDT scores below 70; pink indicates a partial model of high confidence; grey indicates the absence of a model). Domain pLDDT values are given (see also S6 Table). Proteins are identified by their common names or InterPro codes. Organism codes are as follows: TRYB2, *T. brucei*; STYPI, *S. pistillata*; 9CHLO, *Bathycoccus prasinos*; YEAST, *S. cerevisiae*; EMENI, *A. nidulans*; 9GLOM, *R. irregularis*; DICDI, *D. discoideum*; CRYNV, *C. neoformans*; SCHPO, *S. pombe*; TETTS, *T. thermophila*; STRPU, *S. purpuratus*; DROME, *D. melanogaster*; CAEEL, *c. elegans*; 9TELE, *D. translucida*; XENLA, *X. laevis*; AUXPR, *A. protothecoides*; CHLSO, *C. sorokiniana*; ORYSJ, *O. sativa*, subsp. *japonica*; ARATH, *A. thaliana*; TOBAC, *N. tabacum*; KLENI, *K. nitens*; CHABU, *C. braunii*; CERRI, *C. richardii*; MARPO, *M. polymorpha*; PHYPA, *P. patens*; EUKA, *Planoprotostelium fungivorum*; 9FUNG, *Umbelopsis vinacea*; SPOSA, *Sporidiobolus salmonicolor*; CAPO3, *Capsaspora owczarzaki*; 9METZ, *Trichoplax sp. H2*; TRISP, *Trichinella spiralis*; HELRO, *Helobdella robusta*; RAMVA, *Ramazzottius varieornatus*; 9ANNE, *Dimorphilus gyrociliatus*; BUGNE, *Bugula neritina*; SCHHA, *Schistosoma haematobium*; SCHSO, *Schistocephalus solidus*; 9BILA, *A. steineri*; EUMVA, *Eumeta variegata*; 9EUPU, *Arion vulgaris*; LINUN, *Lingula unguis*; ARAVE, *Araneus ventricosus*; AMPQE, *Amphimedon queenslandica*; DANRE, *D. rerio*. (B) Simplified cladogram showing the phylogeny of *REX1* and *REX3* genes. (C) Structural overlay of AF models for Rex3 from *S. cerevisiae* and Q9N5W0 from *C. elegans*. Domains and termini are labelled.

The *half-pint* twin RRM domain was restricted to Rex1 proteins from vertebrates. In contrast to an earlier report [15], the mouse Rexo5 protein has a histidine to arginine substitution within the Exo III motif (Fig 7) [14]. This is not inconsistent with its inferred exonuclease activity, since DEDDh family members with noncanonical catalytic centres have been shown to have inherent exoribonuclease activity [52,53].

The Rex3 clade includes proteins from fungi and the REXO1 proteins from metazoans but lacks proteins from plants or algae. The ancestral *REX3* gene therefore appears to have evolved before the divergence of fungi and metazoans, although a single Rex3 homologue was found in the mycophagus amoeba *Planoprotostelium fungivorum*. The arginine, glutamate and tyrosine residues seen in the characterised KIX domains of CBP and Med15, and the cysteine/histidine-rich signature sequence of CHORD domains (Fig 5), are highly conserved in Rex3-related proteins (S7 Fig). The canonical domain architecture for Rex3-related proteins consists of a tandemly arranged CHORD domain and DEDDh domain at the C-terminus, with an upstream KIX domain. This domain architecture has previously been noted for a CHORD domain protein from the basidiomycete *Rhizopogon vesiculosus* [54]. An exception is the protein from *Trichinella spiralis* and related species, which has two C-terminal thioredoxin domains (S8 Fig). Another unusual architecture is shown by the Rex3 homologue from the rotifer *Adineta steineri*, which has two tandem copies of the DEDDh domain at the C-terminus (S8 Fig), each with an upstream CHORD domain. Both DEDDh domains of the *A. steineri* protein contain canonical active site sequence signatures. Rex1- and Rex3-related proteins typically contain the sequence DCEM within the Exo I motif, the dipeptide DL within the Exo II motif and the dipeptide IQ upstream of the Exo III motif [1]. We noticed a very strong preference for a tyrosine or phenylalanine residue within the Exo I motif of Rex3 homologues (indicated with an asterisk in Fig 7). One possibility is that this hydrophobic residue contributes to substrate binding and/or specificity. Some noncanonical catalytic centres are also observed in the Rex3-related proteins. B8JII5 from *D. rerio* and A0A553MNP1 from *D. translucida* have E/Q substitutions in the Exo I motif, while A0A7J7IZG1 from *Bugula neritina* lacks part of Exo II, including the conserved aspartate residue. Structural overlay of the AF model for the *B. neritina* protein with that of yeast Rex1 suggests the overall fold of the catalytic centre is nevertheless conserved and that an electron pair could potentially be donated by either T840 or S843 (S8 Fig).

Notably, *D. melanogaster* and *Caenorhabditis elegans* express a Rex3 variant that lacks an N-terminal KIX domain (Prage and Q9N5W0_CAEEL, respectively). Prage has an N-terminal region of low structure propensity, whereas a high confidence AF model of the Q9N5W0 protein (pLDDT of 90.4 for the whole length of the protein) suggests a compact domain structure comprising solely of the CHORD and DEDDh domains (Fig 7C). Inspection of the structure similarity clusters within AF suggest that this structural variant is widespread among metazoans and also found in nonpathogenic *Cryptococcus* fungi (S9 Table). Prage is required for turnover of maternally expressed mRNAs during early embryogenesis in flies [55,56]. Some eukaryotes, therefore, encode a functional variant of Rex3 that lacks the KIX domain, in addition to the canonical Rex3 protein.

This study reports structural differences that differentiate members of the Rex1/Rex3 family and provides evidence that characteristic conserved domains and motifs within the *S. cerevisiae* enzymes are required for RNA processing functions *in vivo*. Given the sequence similarity between Rex1 and Rex3 proteins, compared with other DEDDh exoribonucleases, and the restriction of Rex3 to specific eukaryotic lineages, it is probable that the ancestral *REX3* gene evolved from *REX1*. These findings suggest numerous further lines of investigation into mechanistic studies on the structure/function relationship of these enzymes.

## Materials and methods

### Bioinformatics analyses

Search and sequence analyses were carried out using the EMBL-EBI tools services [57]. Initially, DEDDh family exoribonucleases were identified through homology with the DEDDh domain of the yeast family members by iterative PSI–BLAST searches against the UniProtKb swissprot database or retrieved directly from the Ensembl Databank resource [58]. Retrieved sequences were limited to proteins from humans and the model organisms *A. thaliana, C. elegans*, *D. rerio, Dictyostelium discoideum, D. melanogaster, G. gallus, M. musculus, S. cerevisiae, S. pombe* and *X. laevis*. Additional proteins within InterPro entry IPR047021 (RNA exonuclease REXO1/REXO3/REXO4-like) were then selected from the organisms *A. nidulans, A. protothecoides*, *Ceratopteris richardii, C. braunii, C. reinhardtii, C. sorokiniana*, *C. neoformans, D. translucida, K. nitens, M. polymorpha, Nicotiana tabacum, Oryza sativa* subsp*. japonica, Physcomitrium patens, Rhizophagus irregularis*, *S. moellendorffii, Stronglyocentrotus purpuratus, Stylophora pistillata, T. thermophila* and *T. brucei.* Further Rex3-related proteins were retrieved from the InterPro entry IPR031736 (RNA exonuclease 1 homolog-like domain) that also contained an annotated DEDD nuclease domain. Identical sequences were filtered using UniProt Align. Multiple sequence alignments were performed using MAFFT and viewed in JalView [59]. Phylogeny trees were generated using iTOL [60]. AF structural models (Monomer version 4 pipeline) [22,23] within listed structure similarity clusters [28,29] or accessed through InterPro entries were compared manually. pLDDT values for the KIX and CHORD domains, the twin RRM domains and the ß strands of the AlkP domains were determined using the AF webtool.

The following experimentally determined structures were downloaded from PDB [61]: yeast Gal11/Med15 (PDB ID 2K0N) [62]; yeast TFIIS (PDB ID 3PO3) [42]; mouse CBP (PDB ID 4I9O) [63]; human RECQL5 (PDB ID 4BK0) [41]; PUF60 with and without bound RNA (PDB IDs 5KW1, 5KWQ) [33]; *E. coli* alkaline phosphatase (PDB ID 1ALK) [27]; iPGM from *B. stearothermophilus* (PDB ID 1EJJ) [24] and an Hsp90/Sgt1/RAR1 complex (PDB ID 2xcm) [40]. Protein structure overlays were generated in UCSF Chimera [64]. Structure-based multiple sequence alignments were generated in Chimera and viewed in Jalview. AF structures were screened for matches within PDB and vice versa using the Dali server [65].

### Yeast methods

*REX1 and REX3* coding sequences were amplified from yeast genomic DNA using the primers shown in Table 2 and cloned using standard cloning techniques into a plasmid that allows expression in yeast of N-terminal fusion proteins that harbour two copies of the z domain of protein A from *Staphylococcus aureus*. Expression from these plasmids is driven by the *RRP4* promoter [66] (see Table 3). To generate the *rex1* deletion mutants, site-directed mutagenesis was performed on the wild-type expression construct using divergent primer pairs that introduce *Bam*HI sites and Gly/Ser linker sequences at appropriate positions. The *rex1* loop 1Δ mutant (ΔA145-S154) contains the linker GGSGSG, the loop 2Δ mutant (ΔN459-D472) contains the linker GSGSGSGSG and the loop 3Δ mutant (ΔR507-G546) contains the linker GSGSGG. The linker sequences were designed to be of minimal length yet sufficient to span the distance between residues at the beginning and ends of the loops, based on the AF structural model. Constructs were validated by sequence analysis of the complete insert.

**Table 2. Oligonucleotides used in this study.**

| Oligo | Primer/Probe | Sequence | Application | Source |
|---|---|---|---|---|
| o57 | northern probe | AATAGAGGTACCAGGTCAAGAAGC | detection of MRP RNA | [67] |
| o221 | northern probe | GCGTTGTTCATCGATGC | detection of 5.8S rRNA | [13] |
| o242 | northern probe | AAGGACCCAGAACTACCTTG | detection of *SCR1* RNA | [67] |
| o270 | northern probe | TCAGAGATCTTGGTGATAAT | detection of U24 snoRNA | [67] |
| o339 | northern probe | AGAAACAAAGCACTCACGAT | detection of tRNA$^{Arg}_{UCU}$ | [3] |
| o925 | northern probe | CTACTCGGTCAGGCTC | detection of 5S rRNA | [68] |
| o950 | northern probe | CCTTGCTTAAGCAAATGC | detection of pre-tRNA$^{Lys}_{UUU}$ intron | this study |
| o1310 | forward PCR primer | GAAGGATCCGGTTCAGGTGGTGTAGCATCCT | Generation of the *rex1* loop3Δ mutant | this study |
| o1311 | reverse PCR primer | CTTGGATCCTGTGTCGCCTGTACC | Generation of the *rex1* loop3Δ mutant | this study |
| o1312 | forward PCR primer | GAAGGATCCGGTTCAGGAGCGTCCATGGTTCTTC | Generation of the *rex1* loop2Δ mutant | this study |
| o1313 | reverse PCR primer | CTTGGATCCTGAACCAGATCCGCGTGACCTCTCTAAG | Generation of the *rex1* loop2Δ mutant | this study |
| o1314 | forward PCR primer | GGTGGATCCGGATCTGGGCCCTACAATTCATTTATTAATG | Generation of the *rex1* loop1Δ mutant | this study |
| o1315 | reverse PCR primer | CCAGGATCCACCCATCACGGGAAATGTATG | Generation of the *rex1* loop1Δ mutant | this study |
| o1326 | forward PCR primer | GCGATCGATAAACTCTATCACTACAAATC | Generation of the *rex3Δ1–106* allele | this study |
| o1327 | forward PCR primer | GCGATCGATAAGAGGAGAATCATATGAC | Generation of the *rex3Δ1–218* allele | this study |
| o1328 | reverse PCR primer | GCGCTCGAGGTCTTTTAATGCCGCATG | cloning of *rex3* alleles | this study |

**Table 3. Plasmids used in this study.**

| Plasmid | Description | Source |
|---|---|---|
| p674 | wild-type Rex1 expressed as an N-terminal zz protein from the *RRP4* promoter in pRS313. | [18] |
| p966 | loop 2Δ derivative of p674 | this study |
| p967 | loop 1Δ derivative of p674 | this study |
| p972 | loop 3Δ derivative of p674 | this study |
| p973 | wild-type Rex3 expressed as an N-terminal zz protein from the *RRP4* promoter in pRS416. | this study |
| p974 | Δ1–106 variant of p973 | this study |
| p975 | Δ1–218 variant of p973 | this study |

Wild-type (*Mata his3Δ1 met15Δ0 leu2Δ0 ura3Δ0*), *rex1Δ::KANMX4* (*Matα his3Δ1 leu2Δ0 lys2Δ0 ura3Δ0 rex-1Δ::KANMX4*) and *rex3Δ::KANMX4* (*Mata his3Δ1 met15Δ0 leu2Δ0 ura3Δ0 rex3Δ::KANMX4*) strains were obtained from Euroscarf (Frankfurt, Germany). The *rex1Δ rrp47Δ* plasmid shuffle strain (*Mat**a** ade2 ade3 his3 leu2 trp1 ura3 rex-1Δ::KANMX4 rrp47Δ::KANMX4*) harbouring a plasmid encoding the *URA3*, *ADE3* and *RRP47* genes has been reported previously [69]. An *rrp47Δ::HYG* strain was generated by marker replacement of the *rrp47Δ::KANMX4* strain P368 [13] using pAG32 [70]. The resultant strain was crossed twice with a *rex1Δ::KANMX4 ade2⁻* strain [69] to obtain a heterozygous *REX1/rex1Δ::KANMX4 RRP47/rrp47Δ::HYGMX4* diploid strain that exhibited adequate spore viability. The *pop1–1* strain (*Matα ade2 arg4 leu2–3,112 ura3–52 pop1–1*) was generated by backcrossing the original temperature-sensitive isolate [38].

Plasmid transformations were performed using a standard lithium acetate procedure [71]. Yeast transformants were grown in selective SD minimal medium (2% glucose, 0.5% ammonium sulphate, 0.17% yeast nitrogen base, plus appropriate amino acids and bases) lacking uracil or histidine. Strains were grown at 30°C except the *pop1–1* strain, which was harvested during growth at 23°C and after transfer to 37°C for 7 hours. Solid medium contained 2% bacto-agar. Plasmid shuffle assays were carried out on the *rex1Δ rrp47Δ* double mutant after transformation with plasmids encoding the *HIS3*

marker and either wild-type *REX1* or the *rex1* loop mutants. Spot growth assays were performed on ten-fold serial dilutions of saturated cultures of resultant transformants of the plasmid shuffle strain and on three independent FOA-resistant isolates of each *rex1Δ rrp47Δ* double mutant.

The heterozygous *REX1/rex1 RRP47/rrp47* diploid strain was transformed with pRS313-derived plasmids (*HIS3* marker) [72] encoding the wild-type *REX1* fusion protein or loop deletion derivatives and sporulated on solid medium containing 2% potassium acetate and essential amino acids at 30°C for 5–7 days. Resultant tetrads were dissected onto YPD medium using an MSM system microscope (Singer Instruments). Dissection plates were incubated at 30°C for 4 days. 94 meiotic progeny from each strain were randomly selected, grown at 30°C in liquid YPD medium overnight, pinned onto solid media and analysed for growth on YPD medium in the presence or absence of hygromycin and/or G418 sulphate (Melford Laboratories, UK) and on minimal medium lacking histidine. Growth was scored by inspection of the plates after incubation at 30°C for 1–2 days. Candidate *rex1Δ rrp47Δ* double mutants that showed growth on medium containing hygromycin and G418 individually and in combination, and on medium lacking histidine, were screened for expression of plasmid-borne zz-Rex1 fusion proteins by western analysis.

### Protein and RNA analyses

Denatured protein whole cell lysates were prepared under alkaline conditions, as described previously [18]. Western blot analyses were carried out using a peroxidase/antiperoxidase conjugate (P1291, Sigma Aldrich) to detect zz fusion proteins. Pgk1 was detected using a mouse monoclonal antibody (clone 22C5D8, Life Technologies) followed by an HRP-conjugated goat anti-mouse secondary antibody (1706516, Bio-Rad Laboratories). Proteins were visualised by ECL using an iChemi XL GelDoc system fitted with GeneSnap software (SynGene) and quantified using ImageJ (NIH). Rex1 and Rex3 expression levels were determined on three biological replicates, normalised to the expression level of Pgk1 and expressed relative to the level of the wild-type protein.

Total cellular RNA was prepared using a standard guanidinium hydrochloride/phenol extraction method. RNA was resolved through denaturing polyacrylamide/urea gels and transferred to Hybond N$^+$ membranes (GE Healthcare, UK) using standard protocols. Northern hybridisations were carried out using 5'-$^{32}$P-labelled oligonucleotides. $^{32}$P signals were captured using PhosphoImager screens and retrieved using a Typhoon FLA 7000 imager. RNA analyses were performed on two biological replicates.

### Supporting information

**S1 Table. Structure similarity clusters of Rex1-, Rexo5- and Rex3-related proteins.** Proteins were identified in the AF Protein Structure Database using MMseqs2 and Foldseek [22,23]. The datasheets include average pLDDT scores and a link to each model structure. The file also includes the results of searches for structural homologues of the cofactor-independent phosphoglucomutase from *B. stearothermophilus (*PDB ID 1EJJ) in the AF database of *H. sapiens* and *D. melanogaster*.
(XLSX)

**S2 Figure. The AlkP domain of Rex1 proteins from trypanosomes contain an additional extended loop.** Ribbon structures of the AF models of the AlkP domains of Rex1 homologues from *T. brucei brucei* (AF-Q580Z1) and *T. cruzi* (AF-Q4DG67). The perspective shown is towards the surface of the ß-sheet, as shown for proteins in Figure 1. N-terminal sequences are coloured in blue. C-terminal sequences are coloured in red. For clarity, only sequences from strand 1 to strand 2 and from strand 3 to strand 7 are shown.
(PDF)

**S3 Table. Structure similarity between Rexo5 proteins and PUF60/FIR.** Results of structural homology searches within the PDB to identify proteins that share homology with the twin RRM domains of Rexo5 proteins from *H. sapiens*, *M. musculus*, *G. gallus*, *X. laevis* and *D. translucida*. Data from reverse homology searches of human and mouse

species-specific AF databases (version 2) for structural homologues of PUF60/FIR are also included. Searches were carried out using the Dali server [65].
(XLSX)

**S4 Table. Structure homology analyses of Rex3 and REXO1 proteins.** Results of structural homology searches interrogating PDB and AF databases (version 2) with Rex3 residues 17–79 and 113–238 and reverse homology searches using the Gal11 KIX domain (PDB ID 2k0N) and residues 148–221 of RAR1 (PDB ID 2xcm). Searches were carried out using the Dali server [65].
(XLSX)

**S5 Figure. Phylogenetic tree of eukaryotic DEDDh domain proteins.** Unrooted phylogenetic tree of 387 distinct DEDDh domain sequences from diverse eukaryotic organisms. Proteins are labelled and annotated according to UniProt entries. Proteins from *A. thaliana*, *C. elegans*, *S. cerevisiae*, *C. neoformans*, *C. reinhardtii*, *D. discoidium*, *D. rerio*, *D. melanogaster*, chick and humans are indicated in bold type. The positions of major proteins within the tree are indicated.
(PDF)

**S6 Table. Confidence analysis of the AlkP, KIX and CHORD domains.** A summary of the pLDDT scores of AlkP, RRM, KIX and CHORD domains within the Rex1 and Rex3 proteins analysed. Residues constituting each domain within the linked AF models are given. Analysis of the AlkP domain was restricted to strands within the core ß sheet. Values for individual RRMs and the twin RRM domain (from the beginning of ß1 in the N-terminal RRM domain to the end of ß4 in the C-terminal RRM) are given. The pLDDT scores of models with incomplete AlkP domain ß sheets are highlighted in pink.
(XLSX)

**S7 Figure. Sequence alignment of the KIX and CHORD domains of Rex3 proteins.** Structure-based multiple sequence alignment of the KIX (upper panel) and CHORD (lower panel) domains of Rex3-related proteins. Residues are coloured by conservation.
(PDF)

**S8 Figure. Noncanonical Rex3-related proteins.** (A) Unusual domain organisations in Rex3-related proteins. The relative positions of KIX, CHORD (CH), DEDDh and thioredoxin (THX) domains in the proteins from *T. spiralis* and *A. steineri* are indicated. (B) Structural overlay of the catalytic centres of the AF models for yeast Rex3 (blue) and the A0A7J7IZG1 protein from *B. neritina* (brown). The positions of active site residues are shown.
(PDF)

**S9 Table. Rex1-related proteins lacking an N-terminal KIX domain.** Structure similarity clusters based on Q9N5W0 from *C. elegans* and the *D. melanogaster* protein Prage within the AF database were identified using MMseqs2 and Foldseek. Hits containing a KIX domain structure or lacking a CHORD were filtered out manually.
(XLSX)

**S1 raw images . All initial western blot and northern blot data used in this publication.**
(PDF)

## Acknowledgments

P.D. was supported by the White Rose BBSRC doctoral training programme. S.K. and I.T. were on MBiolSci undergraduate programmes within the University of Sheffield. The authors thank Monika Feigenbutz for critical reading of the manuscript.

## Author contributions

**Conceptualization:** Phil Mitchell.

**Formal analysis:** Phil Mitchell.

**Investigation:** Peter W. Daniels, Sophie Kelly, Iwan J. Tebbs.

**Project administration:** Phil Mitchell.

**Supervision:** Peter W. Daniels, Phil Mitchell.

**Writing – original draft:** Phil Mitchell.

**Writing – review & editing:** Phil Mitchell.

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
