## [Decision Letter · Decision Letter 0]

7 Oct 2024

PONE-D-24-32419Conserved domains and structural motifs that differentiate closely related Rex1 and Rex3 DEDDh exoribonucleases are required for their function in yeastPLOS ONE

Dear Dr. Mitchell,

Thank you for submitting your manuscript to PLOS ONE. After careful consideration, we feel that it has merit but does not fully meet PLOS ONE’s publication criteria as it currently stands. Therefore, we invite you to submit a revised version of the manuscript that addresses the points raised during the review process.

The two reviewers provide complementary bioinformatic and molecular biological approaches that, together with my comments, lead to  for major areas that are the highest priority issues prior to resubmission

In my independent review, I was not convinced by the Northern analysis that the authors conclude are indicative of processed and unprocessed tRNA forms. The combination of a lack of resolution and a comparison with predicted mobilities rather than a side by side comparison with precursor and product forms result is inadequate. 

I fully concur with Reviewer 2’s issue on the interpretation of the complementation results in Figure 2D and Figure 3E.  A negative control is required to show that actual complementation is taking place. The heterozygous analysis of meiotic progeny would be the best approach. 

To eliminate gene expression effects of plasmids, it would also be best to integrate the tagged deletion constructs that appear to be used rather than use plasmids. While the plasmid shuffles are highly suggestive, they are not ideal and require assumptions based on published data.

I think that the use of the RRP4 promoter as opposed to using the endogenous promoter is problematic, (although used by the author in published studies) Are these at higher/lower abundance than the endogenous protein> This is essential information to verify the appropriateness of RRP4 promoter use.

As pointed out by Review 1, the AlphaFold data is difficulty to interpret without appropriate referencing, statistical analysis, and a lack of consistency in the presentation of the data.

The authors have many additional comments that must be addressed in a point by point fashion.  

We look forward to receiving your revised manuscript.

Kind regards,

Arthur J. Lustig, PhD

Academic Editor

PLOS ONE

Journal Requirements: When submitting your revision, we need you to address these additional requirements. 1. Please ensure that your manuscript meets PLOS ONE's style requirements, including those for file naming. The PLOS ONE style templates can be found at https://journals.plos.org/plosone/s/file?id=wjVg/PLOSOne_formatting_sample_main_body.pdf and https://journals.plos.org/plosone/s/file?id=ba62/PLOSOne_formatting_sample_title_authors_affiliations.pdf 2. PLOS requires an ORCID iD for the corresponding author in Editorial Manager on papers submitted after December 6th, 2016. Please ensure that you have an ORCID iD and that it is validated in Editorial Manager. To do this, go to ‘Update my Information’ (in the upper left-hand corner of the main menu), and click on the Fetch/Validate link next to the ORCID field. This will take you to the ORCID site and allow you to create a new iD or authenticate a pre-existing iD in Editorial Manager. 3. PLOS ONE now requires that authors provide the original uncropped and unadjusted images underlying all blot or gel results reported in a submission’s figures or Supporting Information files. This policy and the journal’s other requirements for blot/gel reporting and figure preparation are described in detail at https://journals.plos.org/plosone/s/figures#loc-blot-and-gel-reporting-requirements and https://journals.plos.org/plosone/s/figures#loc-preparing-figures-from-image-files. When you submit your revised manuscript, please ensure that your figures adhere fully to these guidelines and provide the original underlying images for all blot or gel data reported in your submission. See the following link for instructions on providing the original image data: https://journals.plos.org/plosone/s/figures#loc-original-images-for-blots-and-gels.   In your cover letter, please note whether your blot/gel image data are in Supporting Information or posted at a public data repository, provide the repository URL if relevant, and provide specific details as to which raw blot/gel images, if any, are not available. Email us at plosone@plos.org if you have any questions.

Reviewers' comments:

Reviewer's Responses to Questions

**Comments to the Author**

1. Is the manuscript technically sound, and do the data support the conclusions?

Reviewer #1: Yes

Reviewer #2: Partly

2. Has the statistical analysis been performed appropriately and rigorously? 

Reviewer #1: Yes

Reviewer #2: N/A

3. Have the authors made all data underlying the findings in their manuscript fully available?

Reviewer #1: No

Reviewer #2: Yes

4. Is the manuscript presented in an intelligible fashion and written in standard English?

Reviewer #1: Yes

Reviewer #2: Yes

5. Review Comments to the Author

Reviewer #1: The manuscript by Daniels, et al., entitled “Conserved domains and structural motifs that differentiate closely related Rex1 and Rex3 DEDDh exoribonucleases are required for the function in yeast” describes phylogenetic and structural analysis of the DEDD family of exonucleases. The authors identify two distinct clades within a subgroup of the DEDD family. They further use mutagenesis approaches to demonstrate that regions outside the DEDD domains play critical roles in in vivo RNA processing events.

The overall conclusions, including the classification of clades with distinct structural features, appears to be well supported. The study provides a useful description of Rex1 and Rex3 exoribonucleases. However, I have concerns with the presentation of the structural data, as described below, which should be addressed before acceptance.

The structural analysis of relies heavily on AlphaFold generated models. The authors should clearly indicate what version of AlphaFold they are using. There are several descriptions of AlphaFold models/structural clusters that aren’t referenced in a figure or supplemental data (for example: pg 7, ln 145; pg 9, ln 214; pg 13, ln 319). Can the predicted models/coordinates be provided as supplemental data and summarized in a figure? Perhaps the final panel in Fig 4A starts to address the question, but it isn’t referenced in relevant places in the main text and the source data is still missing. (Also, please clearly define the confidence thresholds used for the coloring scheme in Fig 4A.) Some AlphaFold models are presented in figures, colored by confidence scores. At other times, the average pLDDT scores for the entire protein are reported (see pg 12, ln 290). I understand the difficulty of succinctly presenting all the data, but there should be an effort to be more consistent. This further underscores the need to make the predicted structures available (with confidence scores).

Figure 1 is extremely difficulty to follow. Panel A: Are all 3 structures AlphaFold models? It seems strange to have different coloring schemes for the structures. The position of the DEDDh with respect to the AlkP fold is difficult to discern. It might help to present the entire structure first, possibly color coded by domain. It would also be helpful to use the same color scheme in the topology diagram. Panel B: Is there a reason the AlkP domain isn’t included in the domain structure figure? What is the relationship between the structures in Panel B and Panel C? (I think the view has simply been rotated and the DEDD removed. Some kind of visual cue would be helpful.) Overall, the idea of a common AlkP fold with a DEDDh insertion is very challenging to understand as presented.

Reviewer #2: The manuscript by Daniels et al. “Conserved domains and structural motifs that differentiate closely related Rex1 and Rex3 DEDD exoribonucleases are required for their function in yeast” addresses the characteristic structural features of Rex1 and Rex3 yeast ribonucleases. The authors employed the Alphafold to predict the high confidence domain structure of the representative proteins from the subgroup with catalytic DEDD domain including the yeast enzymes Rex1 and Rex3, the metazoan REXO1 and Rexo5 proteins, and the plant protein Sdn5. Comparison of protein structure models revealed that this group can be differentiated into two distinct clades - Rex1/Rexo5/Sdn5 and Rex3/REXO. Structural modeling of Rex1 revealed three surface loops directed towards the DEDD domain, one of which forms an extended helical arch that is found in homologues across fungi and plants. Molecular analysis showed that the arch and an adjacent loop are required for Rex1-mediated processing of 5S rRNA and tRNA. Rex3-related proteins contain a KIX domain interacting with CREB kinase-inducible domain and a cysteine- and histidine-rich domain (CHORD) adjacent to a C-terminal DEDD domain. Deletion of the N-terminal region spanning the KIX domain blocked Rex3 function in RNase MRP processing.

Altogether, this work identifies evolutionarily conserved structural hallmarks within Rex1 and Rex3 proteins and demonstrates that specific features are required for Rex1- and Rex3-mediated RNA processing pathways in vivo.

The results are interesting and but publication in PLOS ONE requires some additional work, as specified below.

Critical comments:

1. Figure 2D shows complementation of the synthetic lethal phenotype of double mutant rex1 delta rrp47 delta by rex1 deletion mutants. This experiment lacks a negative control. This is because the plasmid shuffle assay is not a proper approach for complementation study. I would recommend tetrad analysis of the meiotic progeny of heterologous diploid rex1 delta rrp47delta/rex1 delta RRP47 transformed with empty vector and plasmids with REX1 or rex1 deletions.

2. Figure 2C presents Northern blot showing that the molecular phenotypes of deletion mutants varied; loop 2 –delta is like wild type, whereas loop 1 and loop 3 delta are like rex1delta. However, as stated in the abstract only one loop, adjacent to the helical arch, is required for Rex1-mediated 5S rRNA and tRNA processing. Please explain this inconsistence.

3. Why the zzRex1 protein migrates as a lower band in loop3 delta mutant as shown on Figure 2C?

4. Which tag was used for epitope-tagging of Rex1 and Rex3? What “zz” means?

5. Figure 2E: The experiment lacks a negative control What effects of rex3 mutants on SNR8 and SNR30 were expected? Why the level of delta1-218 mutated protein is so low? It desires some comment.

6. How the authors understand the statement “stable RNA processing” which they use several times?

7. The description of the domain structure of Rex3 and presentation on Figure 3A are unclear to me. Text is too long and many short-cuts were used, the idea is lost.

6. PLOS authors have the option to publish the peer review history of their article (what does this mean? ). If published, this will include your full peer review and any attached files.

**Do you want your identity to be public for this peer review?** For information about this choice, including consent withdrawal, please see our Privacy Policy .

Reviewer #1: No

Reviewer #2: No

---

## [Author Response · Author response to Decision Letter 1]

20 Dec 2024

Dear Editor

I am submitting our revised version of the manuscript PONE-D-24-32419 entitled "Conserved domains and structural motifs that differentiate closely related Rex1 and Rex3 DEDDh exoribonucleases are required for their function in yeast". The manuscript has been extensively revised in response to the comments of the reviewers, together with the editor's comments.

Response to Reviewer 1

The manuscript by Daniels, et al., entitled “Conserved domains and structural motifs that differentiate closely related Rex1 and Rex3 DEDDh exoribonucleases are required for the function in yeast” describes phylogenetic and structural analysis of the DEDD family of exonucleases. The authors identify two distinct clades within a subgroup of the DEDD family. They further use mutagenesis approaches to demonstrate that regions outside the DEDD domains play critical roles in in vivo RNA processing events.

The overall conclusions, including the classification of clades with distinct structural features, appears to be well supported. The study provides a useful description of Rex1 and Rex3 exoribonucleases. However, I have concerns with the presentation of the structural data, as described below, which should be addressed before acceptance.

The structural analysis of relies heavily on AlphaFold generated models. The authors should clearly indicate what version of AlphaFold they are using. There are several descriptions of AlphaFold models/structural clusters that aren’t referenced in a figure or supplemental data (for example: pg 7, ln 145; pg 9, ln 214; pg 13, ln 319). Can the predicted models/coordinates be provided as supplemental data and summarized in a figure? Perhaps the final panel in Fig 4A starts to address the question, but it isn’t referenced in relevant places in the main text and the source data is still missing. (Also, please clearly define the confidence thresholds used for the coloring scheme in Fig 4A.) Some AlphaFold models are presented in figures, colored by confidence scores. At other times, the average pLDDT scores for the entire protein are reported (see pg 12, ln 290). I understand the difficulty of succinctly presenting all the data, but there should be an effort to be more consistent. This further underscores the need to make the predicted structures available (with confidence scores).

Figure 1 is extremely difficulty to follow. Panel A: Are all 3 structures AlphaFold models? It seems strange to have different coloring schemes for the structures. The position of the DEDDh with respect to the AlkP fold is difficult to discern. It might help to present the entire structure first, possibly color coded by domain. It would also be helpful to use the same color scheme in the topology diagram. Panel B: Is there a reason the AlkP domain isn’t included in the domain structure figure? What is the relationship between the structures in Panel B and Panel C? (I think the view has simply been rotated and the DEDD removed. Some kind of visual cue would be helpful.) Overall, the idea of a common AlkP fold with a DEDDh insertion is very challenging to understand as presented.

We thank the reviewer for their comments.

The AlphaFold models used were retrieved from the version 4 database. This is now explicitly stated on line 96 and 689.

We have extensively revised the manuscript to include source data that supports structure similarities, as requested by the reviewer. In some cases this is within the expanded set of data shown in the Figures but largely within supplemental Tables and Figures. AlphaFold database accession codes or PDB chain identifiers are included in the Tables. The supplementary Tables include structure comparison Foldseek and MMseqs2 hits for Rex1, Rexo5 and Rex3 (S1 Table), AlphaFold database hits for iPGM (S1 Table), PUF60 RRMs (S3 Table), and the KIX and CHORD domains (S4 Table), as well as PDB hits for Rexo5 RRMs (S3 Table), and the KIX and CHORD domains (S4 Table). S6 Table includes an important structure summary of the AlkP and RRM domains within Rex1 proteins and the KIX and CHORD domains within Rex3 proteins that provides AlphaFold database IDs and InterPro database identifiers, defines the residues within each structural element and provides individual pLDDT scores. The revised figures now include a comparison of AlphaFold models of yeast Rex1 homologues (Fig 3A), the CHORD domains of Rex3, REXO1 and RAR1 (Fig 5C) and AlphaFold models of the AlkP domain of Rex1 homologues in trypanosomes (S2 Figure).

The confidence scores of the AlkP, RRM, KIX and CHORD domains indicated in Fig 4A of the original manuscript (now Fig 7A) are indicated in the figure legend. Moreover, we have included pLDDT scores for each domain. These are also included in the S6 Table, which summarises the domains for each Rex1 and Rex3 homologue analysed.

Within the text, the confidence of a given structure is supported by giving the pLDDT score for either a specific defined number of residues or the full length of the protein. We have provided such confidence cores using a consistent style throughout the revised manuscript.

Structure comparisons of AlphaFold models are presented in Fig 1 and Fig 3 in the revised manuscript in a consistent manner, whereby the models are colour-coded according to the protein domain, as suggested by the reviewer; the DEDDh domain is colured green, the N- and C-terminal regions of the AlkP domain are coloured blue and red, respectively, and the RRMs are coloured gold. The models in the revised Figure 1 are all in the same colour scheme. Structures for iPGM and AlkP are from crystallographic models. This is clearly explained in the figure legend of the revised manuscript and referred to in the text. We have provided an initial view of the complete structure of the protein (Fig 1A), as suggested by the reviewer. The revised colour scheme emphasises the fact that the DEDDh domain is "inserted" into the AlkP domain. We have included the AlkP domain in the domain structure schematic for Rex1-related proteins (Fig 1D), as suggested by the reviewer. We present models of the complete human and fly Rexo5 proteins (lacking the unstructured N-terminal regions) for the reader to be able to compare them with the structure of yeast Rex1. We also show structures of the AlkP domain of the fly Rexo5 protein and the equivalent portion of the human protein that are viewed from the same perspective as that shown for the yeast protein. This is to enable a comparison of the fold of the AlkP domain, which is not clear in the images of the whole protein. We have included PAE maps of the proteins in the revised Fig 1. These maps portray well the interactions between the N- and C-terminal regions of the protein and emphasise that the domain packing within the proteins vary. This point is important, as the precise alignment of the AlkP and DEDDh domains has varying degrees of confidence in the models of the yeast, fly and human proteins, which is explicitly noted in the revised manuscript.

We sincerely thank the reviewer for their comments, which have helped us to significantly improve the manuscript.

Response to Reviewer 2

The manuscript by Daniels et al. “Conserved domains and structural motifs that differentiate closely related Rex1 and Rex3 DEDD exoribonucleases are required for their function in yeast” addresses the characteristic structural features of Rex1 and Rex3 yeast ribonucleases. The authors employed the Alphafold to predict the high confidence domain structure of the representative proteins from the subgroup with catalytic DEDD domain including the yeast enzymes Rex1 and Rex3, the metazoan REXO1 and Rexo5 proteins, and the plant protein Sdn5. Comparison of protein structure models revealed that this group can be differentiated into two distinct clades - Rex1/Rexo5/Sdn5 and Rex3/REXO. Structural modeling of Rex1 revealed three surface loops directed towards the DEDD domain, one of which forms an extended helical arch that is found in homologues across fungi and plants. Molecular analysis showed that the arch and an adjacent loop are required for Rex1-mediated processing of 5S rRNA and tRNA. Rex3-related proteins contain a KIX domain interacting with CREB kinase-inducible domain and a cysteine- and histidine-rich domain (CHORD) adjacent to a C-terminal DEDD domain. Deletion of the N-terminal region spanning the KIX domain blocked Rex3 function in RNase MRP processing.

Altogether, this work identifies evolutionarily conserved structural hallmarks within Rex1 and Rex3 proteins and demonstrates that specific features are required for Rex1- and Rex3-mediated RNA processing pathways in vivo.

The results are interesting and but publication in PLOS ONE requires some additional work, as specified below.

Critical comments:

1. Figure 2D shows complementation of the synthetic lethal phenotype of double mutant rex1 delta rrp47 delta by rex1 deletion mutants. This experiment lacks a negative control. This is because the plasmid shuffle assay is not a proper approach for complementation study. I would recommend tetrad analysis of the meiotic progeny of heterologous diploid rex1 delta rrp47delta/rex1 delta RRP47 transformed with empty vector and plasmids with REX1 or rex1 deletions.

2. Figure 2C presents Northern blot showing that the molecular phenotypes of deletion mutants varied; loop 2 –delta is like wild type, whereas loop 1 and loop 3 delta are like rex1delta. However, as stated in the abstract only one loop, adjacent to the helical arch, is required for Rex1-mediated 5S rRNA and tRNA processing. Please explain this inconsistence.

3. Why the zzRex1 protein migrates as a lower band in loop3 delta mutant as shown on Figure 2C?

4. Which tag was used for epitope-tagging of Rex1 and Rex3? What “zz” means?

5. Figure 2E: The experiment lacks a negative control What effects of rex3 mutants on SNR8 and SNR30 were expected? Why the level of delta1-218 mutated protein is so low? It desires some comment.

6. How the authors understand the statement “stable RNA processing” which they use several times?

7. The description of the domain structure of Rex3 and presentation on Figure 3A are unclear to me. Text is too long and many short-cuts were used, the idea is lost.

We thank the reviewer for their comments.

1. We have included the plasmid shuffle assay data (revised fig 4A) in the revised manuscript. This clearly shows that there is no detectable growth of the plasmid shuffle strain on medium containing FOA when it is transformed with the vector control. As this point was also raised by the editor, we nevertheless carried out meiotic progeny analyses of a heterozygous diploid strain. We were limited to carrying out a random spore analysis rather than tetrad analysis due to the germination efficiency of the available heterozygous diploid strain. We analysed over 80 randomly selected progeny from transformants of the diploid strain harbouring the wild-type Rex1, each loop mutant, the vector control and an untransformed control. The data (Fig 4B, Table 1) agree with published data (Peng et al., 2003) that rex1∆ rrp47∆ double mutants are inviable and support the conclusion that each loop mutant analysed complements this growth phenotype.

2. The abstract states that three surface loops are directed towards the DEDD domain, one of which forms an arch. The abstract then states that the arch and an adjacent loop are required for RNA processing. The text is consistent - the misunderstanding stems from the fact that the arch is one of the loops. We have altered the text slightly to emphasise this. The abstract now reads "The AlkP domain ... contains three surface loops..., one of which forms an extended helical arch... We show that this arch and an adjacent loop are required for Rex1-mediated processing..."

3. The loop 3∆ mutant has the largest deletion, such that the protein is 3-4 kDa less than the wild-type protein and the other mutants and thus migrates faster upon SDS-PAGE analysis. We have made this point explicitly in the revised manuscript (line 336).

4. The zz tag contains two copies of the z domain of Protein A from Staphylococcus aureus. This is explicitly stated in the revised manuscript (line 708).

5. The data in Fig 3E of the original manuscript (Fig 6B of the revised manuscript) shows the expression of RNase MRP RNA in the rex3∆ mutant and in strains expressing either wild-type Rex3 or the deletion mutants. The observation is that the RNase MRP RNA is longer in the rex3∆ mutant and cells expressing the Rex3 mutants than in cells expressing the wild-type Rex3 protein. It is not clear what is meant by an additional negative control, as the rex3∆ mutant is the sample that does not express any Rex3 protein. We have included samples from the pop1-1 temperature-sensitive mutant as a control for the specificity of the northern hybridisation. As predicted, RNase MRP RNA is depleted in the pop1-1 mutant during growth at the nonpermissive temperature. It is not feasible to analyse an RNA from a strain that does not express RNase MRP RNA, as the NME1 gene that encodes RNase MRP RNA is essential for viability.

We have described more clearly in the revised manuscript (p 22) our reasoning for analysing the expression levels of the snoRNAs snR30 and snR8. We present data that Rex3 contains a CHORD domain. These domains form protein interactions with CS domains. The yeast protein Shq1 contains a CS domain and is required for assembly of snoRNAs into RNA/protein complexes. We wanted to test the hypothesis that Rex3 might interact via its CHORD domain with Shq1 and thereby couple RNA processing with assembly of the processed RNA into RNP complexes. There is no clear effect on snR30 or snR8 levels in the rex3∆ mutant but we feel that this negative data is nevertheless worthy reporting.

There are two potential reasons why the level of the rex3 ∆1-218 mutant is markedly reduced. One is that residues 119-142, which are modelled to be packed against the DEDDh domain, are required for its stable expression. The other is that the CHORD domain itself is required for stable protein expression. Characterised CHORD domain proteins function as part of Hsp90 protein chaperone complexes. These explanations are discussed in greater detail in the revised manuscript (p 22).

6. The phrase "stable RNA processing" relates to post-transcriptional RNA processing events that occur on stably expressed RNAs. We have rephrased this in the revised manuscript to be more explicit, referring for example to "the processing of noncoding transcripts to generate functional, stably expressed RNAs" (line 59).

7. The description of Fig 3A (Fig 5A in the revised manuscript) describes the presence of the KIX domain and CHORD domain in Rex3. It outlines homology with characterised proteins containing these domains and describes conservation both in terms of protein structure and sequence homology. The text then discusses the potential functional importance of the KIX domain to Rex3 function in coupling RNA processing to transcription. Finally, the section describes the conservation of KIX and CHORD domains in both Rex3- and REXO1-related proteins. We feel that these are important aspects that need to be covered in the manuscript.

Editors comments

1.

In my independent review, I was not convinced by the Northern analysis that the authors conclude are indicative of processed and unprocessed tRNA forms. The combination of a lack of resolution and a comparison with predicted mobilities rather than a side by side comparison with precursor and product forms result is inadequate.

2. I fully concur with Reviewer 2’s issue on the interpretation of the complementation results in Figure 2D and Figure 3E. A negative control is required to show that actual complementation is taking place. The heterozygous analysis of meiotic progeny would be the best approach.

3. To eliminate gene expression effects of plasmids, it would also be best to integrate the tagged deletion constructs that appear to be used rather than use plasmids. While the plasmid shuffles are highly suggestive, they are not ideal and require assumptions based on published data.

4. I think that the use of the RRP4 promoter as opposed to using the endogenous promoter is problematic, (although

---

## [Decision Letter · Decision Letter 1]

15 Jan 2025

PONE-D-24-32419R1Conserved domains and structural motifs that differentiate closely related Rex1 and Rex3 DEDDh exoribonucleases are required for their function in yeastPLOS ONE

Dear Dr. Mitchell,

Thank you for submitting your manuscript to PLOS ONE. After careful consideration, we feel that it has merit but does not fully meet PLOS ONE’s publication criteria as it currently stands. Therefore, we invite you to submit a revised version of the manuscript that addresses the points raised during the review process.

**Th** e authors have addressed almost all of the issues raised in the critiques.One issue remains however that, if resolved, would add additional rigor to the study. I believe Reviewer 2's concern (*Reviewer 2 was not available for this round* ) regarding a negative control for the deletion strains is a need for proof that there are no other inadvertent defects present in the mutant strain. The loss of a phenotype after transformation with a wild type copy would exclude this possibility and I would recommend highly that additional experiment. Alternatively, if you have already evidence that excludes this possibility, please discuss that data.

We look forward to receiving your revised manuscript.

Kind regards,

Arthur J. Lustig, PhD

Academic Editor

PLOS ONE

Journal Requirements:

Reviewers' comments:

Reviewer's Responses to Questions

**Comments to the Author**

1. If the authors have adequately addressed your comments raised in a previous round of review and you feel that this manuscript is now acceptable for publication, you may indicate that here to bypass the “Comments to the Author” section, enter your conflict of interest statement in the “Confidential to Editor” section, and submit your "Accept" recommendation.

Reviewer #1: All comments have been addressed

2. Is the manuscript technically sound, and do the data support the conclusions?

Reviewer #1: (No Response)

3. Has the statistical analysis been performed appropriately and rigorously? 

Reviewer #1: (No Response)

4. Have the authors made all data underlying the findings in their manuscript fully available?

Reviewer #1: (No Response)

5. Is the manuscript presented in an intelligible fashion and written in standard English?

Reviewer #1: (No Response)

6. Review Comments to the Author

Reviewer #1: (No Response)

7. PLOS authors have the option to publish the peer review history of their article (what does this mean? ). If published, this will include your full peer review and any attached files.

**Do you want your identity to be public for this peer review?** For information about this choice, including consent withdrawal, please see our Privacy Policy .

Reviewer #1: No

---

## [Author Response · Author response to Decision Letter 2]

27 Feb 2025

A single issue with the revised manuscript was raised by the editor that related back to a concern expressed by reviewer 2 in the initial submission. The issue was that a control was not shown in the northern blot analysis of the rex3 mutant in Fig 6 that was felt would add important additional rigour to the study. This oversight on our part was due to a misunderstanding of how the comment was initially phrased. The nature of the issue was clarified through an email exchange with the editor. We feel that we have appropriately and fully addressed this point in the re-revised manuscript.

---

## [Editor Report · Decision Letter 2]

3 Mar 2025

Conserved domains and structural motifs that differentiate closely related Rex1 and Rex3 DEDDh exoribonucleases are required for their function in yeast

PONE-D-24-32419R2

Dear Dr. Mitchell,

We’re pleased to inform you that your manuscript has been judged scientifically suitable for publication and will be formally accepted for publication once it meets all outstanding technical requirements.

Kind regards,

Arthur J. Lustig, PhD

Academic Editor

PLOS ONE
---

## [Editor Report · Acceptance letter]

PONE-D-24-32419R2

PLOS ONE

Dear Dr. Mitchell,

I'm pleased to inform you that your manuscript has been deemed suitable for publication in PLOS ONE. Congratulations! Your manuscript is now being handed over to our production team.

Kind regards,

on behalf of

Dr. Arthur J. Lustig

Academic Editor

PLOS ONE